# Explainability as statistical inference

## Abstract

A wide variety of model explanation approaches have been proposed in recent years, all guided by very different rationales and heuristics. In this paper, we take a new route and cast interpretability as a statistical inference problem. We propose a general deep probabilistic model designed to produce interpretable predictions. The model's parameters can be learned via maximum likelihood, and the method can be adapted to any predictor network architecture, and any type of prediction problem. Our method is a case of amortized interpretability models, where a neural network is used as a selector to allow for fast interpretation at inference time. Several popular interpretability methods are shown to be particular cases of regularized maximum likelihood for our general model. We propose new datasets with ground truth selection which allow for the evaluation of the features importance map. Using these datasets, we show experimentally that using multiple imputation provides more reasonable interpretation.

## 1 Introduction

Fueled by the recent advances in deep learning, machine learning models are becoming omnipresent in society. Their widespread use for decision making or predictions in critical fields leads to a growing need for transparency and interpretability of these methods. While Rudin (2019) argues that we should always favor interpretable models for high-stake decisions, in practice, black-box methods are used due to their superior predictive power. Researchers have proposed a variety of model explanation approaches for black-box models, and we refer to Linardatos et al. (2021) for a recent survey. Finding interpretable methods is hard. The multiplicity of evaluation methods (Afchar et al., 2021; Jethani et al., 2021a; Liu et al., 2021; Hooker et al., 2019) makes it difficult to assess the qualities of the different methods.

In this paper, we will focus on methods that offer an understanding of which features are important for the prediction of a given instance. These types of methods are called instance-wise feature selection and quantify how much a prediction change when only a subset of features is shown to the model. Ribeiro et al. (2016) fit an interpretable linear regression locally around a given instance, and Seo et al. (2018) create saliency maps by back-propagating gradient through the model. Lundberg & Lee (2017) approximate Shapley values (Shapley, 1953) for every instance. Fong & Vedaldi (2017), Slack et al. (2020) propose to create a saliency map by evaluating the change in performance of the model when exposed to different selections of features. In practice, these methods show very good results but focus on local explanations for a single instance. An evaluation of the selection for a single image requires an exponential number of passes through the black-box model.

It is of particular interest to obtain explanations of multiple instances using amortized explanation methods. The idea of such methods is to train a *selector* network that will generalize the selection given for a single instance. While there is a higher cost of entry due to training an extra network, the interpretation at test time is much faster. Jethani et al. (2021b) proposed to obtain Shapley values with a selector network. Chen et al. (2018), Yoon et al. (2018) both proposed to train a selector that selects a minimum subset of features while maximizing an information-theoretical threshold. Jethani et al. (2021a) showed that the selector in such models was not really constrained to select the important features, but it would encode target information to facilitate the prediction, and proposed a method that trains a surrogate predictor to alleviate this issue.

In this paper, we propose **LEX (Latent Variable as Explanation)** a modular self-interpretable probabilistic model class that allows for instance-wise feature selection. LEX is composed of three

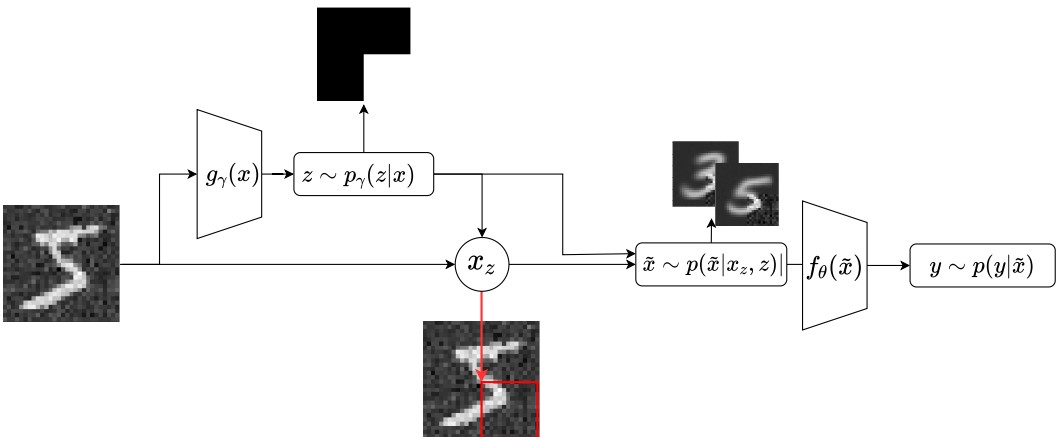

Figure 1: The LEX pipeline allows us to transform any prediction model into an explainable one. In supervised learning, a standard approach uses a function $f_\theta$ (usually a neural network) to parameterize a prediction distribution $p_\theta$. In that framework, we would feed the input data directly to the neural network $f_\theta$. Within the LEX pipeline, we obtain a distribution of masks $p_\gamma$ parameterized by a neural network $g_\gamma$ from the input data. Samples from this mask distribution are applied to the original image $x$ to produce incomplete samples $x_z$. We implicitly create the missingness by sampling imputed samples $\tilde{x}$ given the masked image using a generative model conditioned on both the mask and the original image. These samples are then fed to a classifier $f_\theta$ to obtain a prediction. As opposed to previous methods, multiple imputation allows us to minimise the encoding happening in the mask and to get a more faithful selection.

different modules: a predictor, a selector, and an imputation scheme. We show that up to different optimization procedures, other existing amortized explanation methods (L2X Chen et al. (2018), Invase Yoon et al. (2018), and REAL-X Jethani et al. (2021a)) optimize an objective that can be framed as the maximization of a LEX model. LEX can be used either "In-Situ," where the selector and predictor are trained jointly, or "Post-Hoc," to explain an already learned predictor. We propose two new datasets to evaluate the performance of instance wise feature selection and experimentally show that using multiple imputation leads to more plausible selection.

**Notation**  Random variables are capitalized, their realizations are not. Exponents correspond to the index of realisations and indices correspond to the considered feature. For instance, $x_j^i$ corresponds to the $i^{\text{th}}$ realization of the random variable $X_j$, which is the $j^{\text{th}}$ feature of the random variable $X$. Let $i \in [\![0, D]\!]$, $x_{-i}$ is defined as the vector $(x_0, \ldots, x_{i-1}, x_{i+1}, \ldots, x_D)$, *i.e.*, the vector with $i^{\text{th}}$ dimension removed. Let $z \in \{0,1\}^D$, then $X_z$ is defined as the vector $(x_j)_{\{j|z_j=1\}}$ where we only select the dimensions where $z = 1$, and $x_{1-z}$ denotes the vector $(x_j)_{\{j|z_j=0\}}$ where we only select the dimension where $z = 0$. In particular, $X_z$ is $\|z\|$-dimensional and $x_{1-z}$ is $(D - \|z\|)$-dimensional.

## 2 CASTING INTERPRETABILITY AS STATISTICAL LEARNING

Let $\mathcal{X} = \prod_{d=1}^{D} \mathcal{X}_i$ be a $D$-dimensional feature space and $\mathcal{Y}$ be the target space. We consider two random variables $\mathbf{X} = (X_1, \ldots, X_D)$ and $Y \in \mathcal{Y}$ following the true data generating distribution $p_{\text{data}}(x, y)$. We have access to $N$ i.i.d. realisations of these two random variables, $x^1, \ldots, x^N \in \mathcal{X}$ and labels $y^1, \ldots, y^n \in \mathcal{Y}$. We want to approximate the conditional distribution of the labels $p_{\text{data}}(y|x)$ and discover which subset of features are useful for every local prediction.

### 2.1 STARTING WITH A STANDARD PREDICTIVE MODEL

To approximate this conditional distribution, a standard approach would be to consider a predictive model $\Phi(y|f_\theta(x))$, where $f_\theta : \mathbb{R}^D \to H$ is a neural network and $(\Phi(\cdot|\eta))_{\eta \in H}$ is a parametric family of densities over the target space, here parameterized by the output of $f_\theta$. Usually, $\Phi$ is a categorical distribution for a classification task and a normal distribution for a regression task. The model being

posited, various approaches exist to find the optimal parameters such as maximum likelihood or Bayesian inference. This method is just a description of the usual setting in deep learning. This is the starting point for our models, from which we will derive a latent variable as explanation model.

## 2.2 LATENT VARIABLE AS EXPLANATION (LEX)

As discussed in Section 1, the prediction model $\Phi(y|f_\theta(x))$ is not interpretable by itself in general. Our goal is to embed it within a general interpretable probabilistic model. In addition, we want this explanation to be easily understandable by a human, thus we propose to have a score per feature defining the importance of this feature for prediction. We propose to create a latent $Z \in \{0,1\}^D$ that corresponds to a subset of selected features. The idea is that if $Z_d = 1$, then feature $d$ is used by the predictor, and conversely.

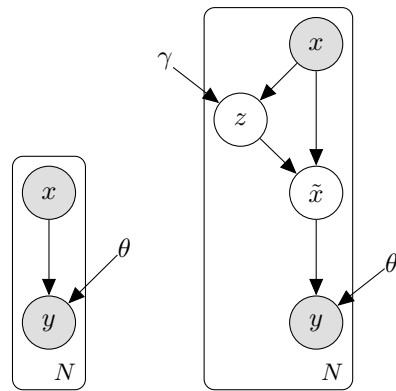

We endow this latent variable with a distribution $p_\gamma(z|x)$. This distribution $p_\gamma$ is parametrized by a neural network $g_\gamma : \mathcal{X} \to [0,1]^D$ called the *selector* with weights $\gamma \in \Gamma$. To obtain the importance feature map of an input $x$, we look at its average selection $\mathbb{E}_\gamma[Z|x]$ or an estimate if not readily available (see Fig. 7). A common parametrization is choosing the latent variable $Z$ to be distributed as a product of independent Bernoulli variables $p_\gamma(z|x) = \prod_{d=1}^{D} \mathcal{B}(z_d|g_\gamma(x)_d)$. With that parametrization, the im-

Figure 2: *Left panel:* graphical model of a standard predictive model. We propose to embed this model in a latent explainer model using the construction of the *right panel.*

portance feature map of an input $x$ is directly given by the output of the selector $g_\gamma$. For instance, Yoon et al. (2018) and Jethani et al. (2021a) use a parametrization with independent Bernoulli variables and obtain the feature importance map directly from $g_\gamma$. L2X (Chen et al., 2018) uses a neural network $g_\gamma$ called the selector but since the parametrization of $p_\gamma$ is not an independent Bernoulli, they obtain their importance feature map by ranking the features' importance with the weights of the output of $g_\gamma$. FastSHAP (Jethani et al., 2021b) also uses a similar network $g_\gamma$ but this network's output holds a different meaning. Indeed, it outputs the Shapley value of the feature for a given instance, hence framing a different notion of interpretability.

In the next sub-section, we will define how feature turn-off should be incorporated in the model, *i.e.*, we will define $p_\theta(y|x,z)$, the predictive distribution given that some features are ignored. With these model assumptions, the predictive distribution will be the average over likely interpretations,

$$p_{\theta,\gamma}(y|x) = \sum_{z \in \{0,1\}^D} p_\theta(y|x,z)p_\gamma(z|x).\tag{1}$$

## 2.3 TURNING OFF FEATURES

Turning off features for a given predictor is a very common problem in interpretable AI (Chang et al., 2019; Frye et al., 2021; Aas et al., 2021; Covert et al., 2021; Rong et al., 2022). We want our explanation to be model-agnostic to embed any kind of predictor into a latent explainer. To that end, we want to make use of $f_\theta$ the same way we would in the setting without selection in Section 2.1, with the same input dimension. Hence we implicitly turn off features by considering an imputed vector $\tilde{X}$. The imputed vector takes the same values as $x$ for $z = 1$ and follows an imputing scheme when $z = 0$. Given $x$ and $z$, $\tilde{X}$ is defined by the following generative process:

1. We sample the imputation part according to the imputing scheme $\hat{X} \sim p_\iota(\hat{X}_{1-z}|x_z)$.

2. Then, we define $\tilde{X}_j = x_j$ if $z_j = 1$ and $\hat{X}_j$ if $z_j = 0$.

For instance, a simple imputing scheme is constant imputation, *i.e.*, $\tilde{X}$ is put to $c \in \mathbb{R}$ whenever $z_j = 0$. Thus, 0-imputation corresponds to replacing the missing features with 0. One could also use the true conditional imputation $p_\iota(\hat{X}_{1-z}|x_z) = p_{\text{data}}(X_{1-z}|x_z)$. Note that constant imputation

is a case of single imputation, meaning that for a given mask $z$ and incomplete instance $x_z$, the reconstruction $\tilde{x}$ is entirely determined. On the contrary, multiple imputation allows for multiple options for reconstruction. Denoting the density for the generative process above $p(\tilde{x}|x, z)$, we define predictive distribution given that some features are ignored as

$$p_\theta(y|x, z) = \mathbb{E}_{\tilde{X} \sim p(\tilde{X}|x,z)} \Phi(y|f_\theta(\tilde{X})) = \mathbb{E}_{\tilde{X} \sim p(\tilde{X}|x,z)} p_\theta(y|\tilde{X}) \,. \tag{2}$$

Fig. 1 shows how the model unfolds. This construction allows us to define the following factorisation of the complete model in Fig. 2:

$$p_{\gamma,\theta}(y, x, \tilde{x}, z) = p_\theta(y|\tilde{x}) p_\iota(\tilde{x}|z, x) p_\gamma(z|x) p(x) \,. \tag{3}$$

## 2.4 STATISTICAL INFERENCE WITH LEX

Now that LEX is cast as a statistical model, it is natural to infer the parameters using maximum likelihood estimation. The log-likelihood function is

$$\mathcal{L}(\theta, \gamma) = \sum_{n=1}^{N} \log p_{\theta,\gamma}(y|x) = \sum_{n=1}^{N} \log[\mathbb{E}_{Z \sim p_\gamma(\cdot|x^n)} \mathbb{E}_{\tilde{X} \sim p(\cdot|(x^n)Z)} p_\theta(y^n|\tilde{X})] \,. \tag{4}$$

Maximizing the previous display is quite challenging since we have to optimize the parameters of an expectation over a discrete space inside a log. Leveraging discrete latent variable model literature Appendix E, we are able to formulate good estimates of Eq. equation 4 gradients and optimize the model using stochastic gradient descent.

If $p_\gamma(z|x)$ can be any conditional distribution, then the model can just refuse to learn something explainable by setting $p_\gamma(z|x) = \mathbf{1}_{z=1_d}$. All features are always turned on. Experimentally, it appears that some implicit regularization or differences the in initialisation of the neural networks may prevent the model from selecting everything. Yet without any regularization, we leave this possibility to chance. Note that this regularization problem appears in other model explanation approaches. For instance, LIME (Ribeiro et al., 2016) fits a linear regression locally around a given instance we want to explain. For the linear regression to be interpretable, they add a regularization as a measure of complexity for this model.

The first type of constraint we can add is an explicit function-based regularization $R : \{0, 1\}^D \to \mathbb{R}$. This function is to be strictly increasing with respect to inclusion of the mask so that the model reaches a trade-off between selection and prediction score. The regularization strength is then controlled by a positive hyperparameter $\lambda$. For instance, Yoon et al. (2018) proposed to used an L1-regularization on the average selection map $R(z) = \|z\|$. While this allows for a varying number of feature selected per instance, the optimal $\lambda$ is difficult to find in practice. Another method considered by Chen et al. (2018) is to enforce the selection within the parametrization of $p_\gamma$. Indeed, they only consider distributions such that any sampled subset have a fixed number of selected features $k$, i.e., $\sum_{d=1}^{D} z_d = k$. This work was extended by Xie & Ermon (2019) who proposed a continuous relaxation of weighted reservoir sampling.

## 2.5 LEX IS ALL AROUND

LEX is a modular framework for which we can compare elements for each of the parametrizations:

- the distribution family and parametrization for the predictor $p_\theta$;
- the distribution family and parametrization for the selector $p_\gamma$;
- the regularization function $R : \{0, 1\}^D \to \mathbb{R}$ to ensure some selection happens. This regularization can be implicit in the distribution family of the selector;
- the imputation function $p_\iota$, probabilistic or deterministic, that handles feature turn-off.

By choosing different combinations, we can obtain a model that fits our framework and express interpretability as the following maximum likelihood problem:

$$\max_{\theta,\gamma} \sum_{n=1}^{N} \left[ \log \mathbb{E}_{Z \sim p_\gamma(\cdot|x^n)} \mathbb{E}_{\tilde{X} \sim p(\cdot|x^n, Z)} p_\theta(y^n|\tilde{X}) - \lambda \mathbb{E}_{Z \sim p_\gamma(\cdot|x^n)} [R(Z)] \right] \,. \tag{5}$$

Table 1: Existing models and their parametrization in the LEX framework.

| Model | Sampling ($p_\gamma$) | Regularization ($R$) | Imputation ($p_\iota$) | Training regime |
|---|---|---|---|---|
| L2X | Subset sampling | Implicit in $p_\gamma$ | 0 imputation | Surrogate PostHoc |
| Invase | Bernoulli | L1-regularization | 0 imputation | InSitu |
| REAL-X | Bernoulli | L1-regularization | Surrogate 0 imputation | Fixed $\theta$ InSitu / Surrogate PostHoc |

Many models, though starting from different formulations of interpretability, minimize a cost function that is a hidden maximum likelihood estimation of a parametrization that fits our framework. Indeed, L2X (Chen et al., 2018) frames their objective from a mutual information point of view, Invase (Yoon et al., 2018) (and REAL-X (Jethani et al., 2021a) whose objective is derived from Invase's objective) frame their objective from a Kullback-Leibler divergence but we can cast the optimization as the maximization of the log-likelihood of a LEX model. We refer to Table 1 for an overview, and to Appendix A for more details.

### 2.6 LEX UNIFIES POST-HOC AND IN-SITU INTERPRETATIONS

While our model can be trained from scratch as a self-interpretable model (we call this setting In-Situ), we can also use it to explain a given black-box model in the same statistical learning framework (this setting is called Post-Hoc). In the In-Situ regime, the selection learns the minimum amount of features to get a prediction closer to the true output of the data while in the Post-hoc regime, the selection learns the minimum amount of features to recreate the prediction of the fully observed input of a given classifier $p_m(y|x)$ to explain. The distinction between the "In-Situ" regime and the "Post-hoc" regime is mentioned, for instance, in Watson & Floridi (2021) as crucial. We distinguish four types of regimes:

**Free In-Situ regime** training an interpretable model from scratch using the random variable $Y \sim p_{\text{data}}$ as a target.

**Fix-$\theta$ In-Situ regime** training only the selection part of the interpretable model using a fixed classifier but still using the random variable $Y \sim p_{\text{data}}(Y|X)$ as a target. In that setting, we do not get an explanation of how $p_\theta$ predict its output but an explanation map for the full LEX model $p_{\theta,\gamma}$.

**Self Post-Hoc regime** training only the selection part of the model using a fixed classifier $p_\theta$, but the target is given by the output of the same fixed classifier when using the full information $Y \sim \Phi(\cdot|f_\theta(x))$. This can be understood as a Fix-$\theta$ In-Situ regime where the dataset is generated from $p_{\text{data}}(x)p_\theta(y|x)$.

**Surrogate Post-Hoc regime** training both the selection and the classification part but the target $Y$ is following the distribution of a given fixed classifier $p_m(y|x)$. The full model $p_{\theta,\gamma}$ is trained to mimic the behaviour of the model $p_m(y|x)$. This can be understood as a Free In-Situ regime where the dataset is generated from $p_{\text{data}}(x)p_m(y|x)$.

## 3 HOW TO TURN OFF FEATURES?

We want to create an interpretable model where an un-selected variable is not "seen" by the classifier of the model. For instance, for a given selection set $z \in \{0,1\}^D$ and a given predictor $p_\theta$, a variable is not "seen" by the predictor when averaging all the possible outputs by the model over the unobserved input given the observed part for a given data distribution $p_\iota$. This allows us to define the restricted predictor with respect to the distribution $p_\iota$:

$$p_{p_\iota}(y|x_z) = \int_{x_{1-z}} p(y|x_{1-z}, x_z)p_\iota(x_{1-z}|x_z)\mathrm{d}x_{1-z}. \tag{6}$$

Covert et al. (2021) advocates the use of the true conditional imputation $p_{\text{data}}(x_{1-z}|x_z)$ for $p_\iota$. This motivates the use of a multiple imputation schemes that should mimic the behaviour of the true conditional imputation $p_\iota = p_{\text{data}}$ for our model.

Chen et al. (2018) and Yoon et al. (2018) proposed to use $0$ imputation in their parametrization as a very rough approximation of the true conditional imputation. Using such imputation can lead to issues. Notably, Jethani et al. (2021a) showed that, with such an imputation, training the selector and predictor jointly can lead to the selector encoding the output target distribution for the classifier making the interpretation incorrect. Instead of using an imputation scheme that would approximate the true conditional imputation, Jethani et al. (2021a) proposed to separate the optimization of $\theta$ and $\gamma$. $p_\theta$ is first trained to approximate the restricted predictor $p_{\text{data}}(y|x_z)$ for any $z \in \{0, 1\}$. Then, the selector part of the model $p_\gamma$ is optimized using a fixed $\theta$. If $p_\theta(y|x, z)$ approximates correctly $p_{\text{data}}(y|x_z)$, the selector should not encode the output of the prediction.

However, training $p_\theta(y|x, z)$ to approximate the true conditional imputation for any mask pattern $z$ is very complicated. Indeed, Le Morvan et al. (2021) showed that the optimal function with single imputation data suffers from discontinuities which makes it a very difficult estimation problem. If the model is not correctly approximating $p_{\text{data}}(y|x_z)$, there is no guarantee that the selection will be meaningful. Jethani et al. (2021a) advocates the use of a constant that is outside the support. Yet, all the experiments are made using $0$ imputation which is inside the support of the input distribution. Having a constant imputation inside the domain of the input distribution may lead to further discrepancy between the surrogate and the restricted predictor. Indeed, Ipsen et al. (2022) showed that using constant imputation inside the domain to learn a purely discriminative models with missing data leads to some artefacts in the prediction.

On the other hand, we propose to approximate the true conditional distribution by using a multiple imputation scheme. This generative model should allow for fast sampling of the quantity $p_\iota(\tilde{x}|x, z)$. Depending on the difficulty of the dataset, obtaining an imputing scheme allowing for fast and efficient masked imputation can be complicated. We show that we can come up with simpler imputing schemes that perform as well or better than the fixed constant ones. For instance, we propose to train a mixture of Diagonal Gaussian to sample the missing values or to randomly sample instances from the validation dataset and replace the missing features with values from the validation sample. We describe many different methods for imputing with multiple imputation in Appendix B.

## 4 EXPERIMENTS

There are many different sets of parameters to choose from in the LEX model. In this section, we want to study the influence of the imputation method in the LEX models as this was the discussion of previous recent papers (Covert et al., 2021; Jethani et al., 2021a). We show the benefits of multiple imputation and that surrogate constant imputation can still lead to encode the target output in the masks.

For all these existing methods, we fix the Monte-Carlo gradient estimator to REBAR Tucker et al. (2019) as it provides an unbiased and low variance estimation of the gradients. We use an implicit regularization constraining the distribution $p_\gamma$ to sample a fixed proportion of feature (note that this is a very similar setting as the one considered in L2X except for the Monte-Carlo gradient estimator). We call this proportion the selection rate. This choice of regularization allows to compare all sets of parameters on the same "credit" for selection (see F.2 for more details)

Evaluation of features' importance map is hard. These evaluation methods rely very often on synthetic datasets where a ground truth is available (Liu et al., 2021; Afchar et al., 2021) or retrain a full model using the selected variable (Hooker et al., 2019). We consider more complicated datasets than these synthetic datasets. We create three datasets using MNIST, FashionMNIST, and CelebA as starting points. These datasets provide information only on a subset of features that should not be selected since it is not informative for the prediction task. It allows us to consider any selection outside this remaining subset as an error in selection. We call this type of selection ground truth the maximum selection ground truth. Since we want to evaluate the selection on ground truths from the dataset, all our experiments are made in the In-Situ regime.

To compare our selections to these ground truths, we look at the true positive rate (TPR), false positive rate (FPR), and false discovery rate (FDR) of the selection map. To that end, we sample at test time a 100 mask samples from $p_\gamma$, and we calculate the three measures for each one of the 100 masks. We then show the average of the selection performance over these 100 masks. To compare the prediction performance, we consider the accuracy of the output averaged over the mask samples.

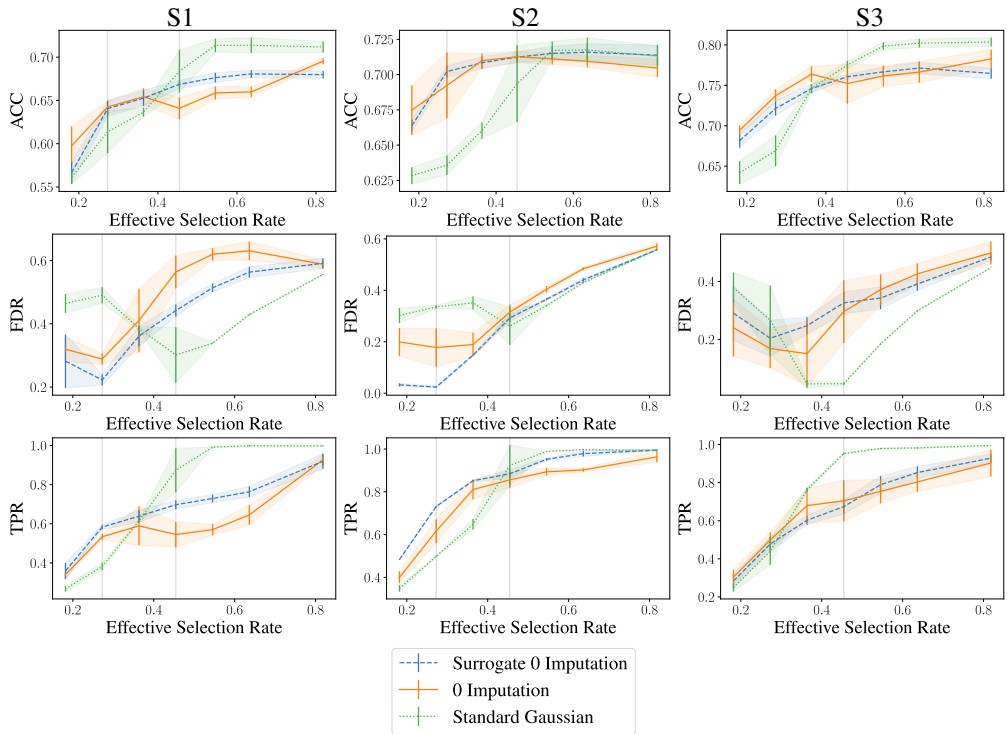

Figure 3: Performances of LEX with different imputations. 0 imputation (solid orange line) corresponds to the imputation method of Invase/L2X, Surrogate 0 imputation (blue dashed line) is the imputation method of REAL-X. The standard Gaussian is the true conditional imputation method from the model (green dotted curve). Columns correspond to the three synthetic datasets (S1, S2, S3) and lines correspond to the different measure of quality of the model (Accuracy, FDR, TPR). We report the mean and the standard deviation over 5 folds/generated datasets.

## 4.1 ARTIFICIAL DATASETS

**Datasets** We study 3 synthetic datasets (S1, S2, S3) proposed by Chen et al. (2018) where the features are generated with standard Gaussian. Depending on the sign of the control flow feature (feature 11), the target will be generated according to a Bernoulli distribution parameterized by one among the three functions described in Appendix C. These functions use a different number of features to generate the target. Thus, depending on the sign of the control flow feature, S1 and S2 either need 3 or 5 features to fully generate the target while S3 always requires 5.

For each dataset, we generate 5 different datasets containing each 10,000 train samples and 10,000 test samples. For every dataset, we train different types of imputation with a selection rate in $[\frac{2}{11}, \frac{3}{11}, \ldots, \frac{9}{11}]$, *i.e.*, we select $2, \ldots, 9$ features. We then report the results in Fig. 3 with their standard deviation across each 5 generated datasets. In the following experiments, we compare a multiple imputation based on a standard Gaussian and constant imputation with and without surrogate using a constant (as used by Yoon et al., 2018; Chen et al., 2018; Jethani et al., 2021a).

**Selection evaluation** The ground truth selection for S1 and S2 have two different true selection rates depending on the sign of the $11^{th}$ feature (shown as two grey lines in Fig. 3). In Fig. 3, for S1, using multiple imputations outperforms other methods as soon as the number of selected variables approaches the larger of the two true selection rates. For S2, LEX performs better most of the time when the selection rate is higher than the larger of the two true selection rates but has a higher variance in performance. LEX outperforms both a 0-imputation and a surrogate with 0-imputation on S3 as soon as the selection rate is close to the true selection rate (shown as the grey line) while still maintaining a better accuracy.

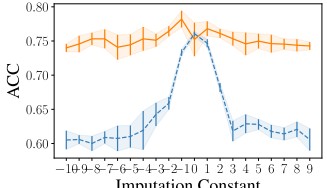 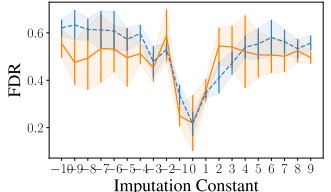 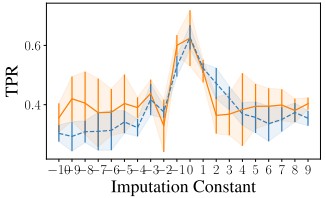

Figure 4: Performances of LEX with varying constant imputation (orange solid line) and surrogate constant imputation (blue dashed line) on S3 using the true selection rate for the selection distribution. Though Invase/L2X (resp. REAL-X) uses constant imputation (resp. surrogate constant imputation), all these methods used only 0 as imputation constant. We report the mean and standard deviation over 5 folds/generated datasets.

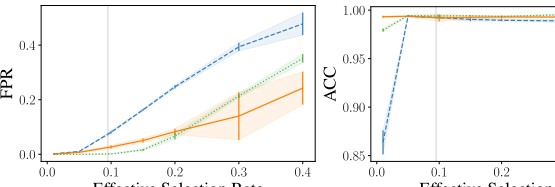 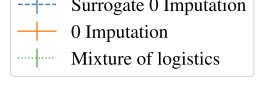

Figure 5: Performances of LEX on the Switching Panel MNIST dataset with different imputations. Surrogate 0 imputation corresponds to the parametrization of REAL-X, 0 imputation corresponds to the parametrization of Invase/L2X We report the mean and standard deviation over 5 folds/generated datasets.

**Dependency on the constant of imputation** We now focus on S3 and we consider the same experiment with 5 generated datasets but with a fixed selection rate $\frac{5}{11}$ corresponding to the true number of features, $k = 5$. We train a single model for both the constant imputation with and without a surrogate varying the constant from $-10$ to $9$. In Fig. 4, the quality of both the selection and the accuracy depends on the value of the imputation constant. Both the surrogate constant imputation and constant imputation perform better when the constant is within the domain of the input distribution. The performance drops drastically outside the range $[-1, 1]$. Jethani et al. (2021a) suggested using a constant outside the domain of imputation which is clearly sub-optimal. Further results in Fig. 10 explicit this dependency on all three synthetic datasets.

## 4.2 SWITCHING PANELS DATASETS

We want to use more complex datasets than the synthetic ones while still keeping some ground truth explanations. To create such a dataset, we randomly sample a single image from both MNIST and FashionMNIST (Xiao et al., 2017) and arrange them in a random order to create a new image. The target output will be given by the target of MNIST for Switching panels MNIST (SP-MNIST) and by the target of FashionMNIST for Switching panels Fashion-MNIST (SP-FashionMNIST).

Given enough information from the panel from which the target is generated, the classifier should not need to see any information from the second panel. If a selection uses the panel from the dataset that is not the target dataset, it means that the predictor is leveraging information from the mask. We consider that the maximum ground truth selection is the panel corresponding to the target image.

Figure 6: Samples from SP-MNIST and their LEX explanations. The class of the top sample is 0, the class of the bottom sample is 4.

We train every model on 45,000 images randomly selected on the train dataset from MNIST (the remaining 15,000 images are used in the validation dataset). For each model, we make 5 runs of the model on 5 generated datasets (*i.e.* for each generated dataset, the panels are redistributed at random) and report the evaluation of each model on the same test dataset with their standard deviation over the 5 folds. All method uses a U-NET (Ronneberger et al., 2015) for the selection and a fully convolutional neural network for the classification. Note that in practice for MNIST, only 19% of pixels are lit on average per image. The true minimum effective rate for our image should therefore

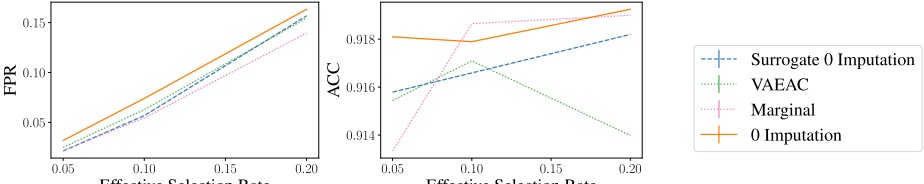

Figure 8: Performances of LEX on the CelebA Smile dataset with different methods of approximation for the true conditional distribution.

be around $10\%$ of the pixels. We evaluate the results of our model around that reasonable estimate of the effective rate. The selected pixels around this selection rate should still be within the ground truth panel.

In Fig. 5, we can see that LEX using multiple imputation with a mixture of logistics to approximate the true conditional imputation outperforms both using a constant imputation (L2X/Invase) and a surrogate constant imputation (REAL-X). Indeed, around the estimated true selection rate $(10\%)$, less selection occurs in the incorrect panel while still maintaining high predictive performance. Fig. 6 shows that LEX uses information from the correct panel of SP-MNIST. We highlight further results and drawbacks in Appendix F.

### 4.3 CELEBA SMILE

The CelebA dataset (Liu et al., 2018) is a large-scale face attribute dataset with more than $200,000$ images which provides $40$ attributes and landmarks or positions of important features in the dataset. Using these landmarks, we are able to create a dataset with a minimal ground truth selection. The assignment associated with the CelebA smile dataset is a classification task to predict whether a person is smiling. We leverage the use of the landmark mouth associated with the two extremities of the mouth to create a ground truth selection in a box located around the mouth. We make the hypothesis that the model should look at this region to correctly classify if the face is smiling or not in the picture (see Appendix C and Fig. 9 for details and examples).

In Fig. 8, we evaluate two methods of multiple imputation. The VAEAC Ivanov et al. (2018) which is a generative model allowing for the generation of imputed samples given any conditional mask $z$ and a marginal multiple imputation. Both our multiple imputation methods perform similarly compared to the constant imputation method both in selection and accuracy. In Fig. 7, we see that LEX uses the information around the mouth to predict whether the face is smiling or not which is a sensible selection. (See Appendix F for further results and samples)

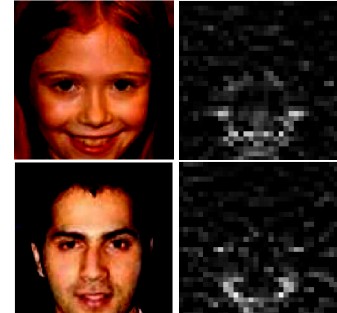

Figure 7: Samples from CelebA smile and their LEX explanations with a $10\%$ rate. Using constant imputation leads to less visually compelling results (Appendix F).

## 5 CONCLUSION

We proposed a framework, LEX, casting interpretability as a maximum likelihood problem. We have shown that LEX encompasses several existing models. We provided 2 datasets on complex data with a ground truth to evaluate the feature importance map. Using these, we compared many imputation methods to remove features and showed experimentally the advantages of multiple imputation compared to other constant imputation methods. These new datasets can be used for other explanation methods than the amortized ones we focused on here.

The framing of interpretability as a statistical learning problem allowed us to use maximum likelihood to find optimal parameters. One could explore other methods for optimizing the parameters, such as Bayesian inference. Interpretation maps are more easily readable when they provide smooth segmentations for images. Another avenue for future work could be to study the use of different parametrizations or regularization that favors connected masks (in the sense that neighbouring pixels have more incentives to share the same selection) to allow for smoother interpretability maps.

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

## A    OTHER METHODS FALLING INTO THE LEX FRAMEWORK

In this section, we show how to cast three existing models in our statistical learning setting. Though the original papers framed their wanted interpretabiliy through different points of view, we will show that what they actually optimize can be seen as *likelihood lower bounds of specific LEX models*.

### A.1    LEARNING TO EXPLAIN (L2X, CHEN ET AL., 2018)

Let us consider a fixed classifier $p_m$. For a fixed number of features $k$, we want to train a selector $p_\gamma$ (this corresponds to $V(\theta, \cdot)$ in the original paper) that will allow selection for this exact number of features (using subset sampling) only and a predictor $p_\theta$ (this corresponds to $g_\alpha$ in the original paper) such that the full model $p_{\theta,\gamma}$ will mimic the behaviour of $p_m$. We use 0-imputation to parameterize the imputation distribution. We can rewrite Eq. (6) in Chen et al. (2018) using our notation as

$$
\mathbb{E}_{X \sim p_{\text{data}}, z \sim p_\gamma} \left[ \sum_y p_m(y|x) \log p_\gamma(z|x) p_\theta(y|x, z) \right]
$$

$$
= \mathbb{E}_{X \sim p_{\text{data}}, z \sim p_\gamma} \left[ \sum_y p_m(y|x) \log p_\gamma(z|x) p_\theta(y|z \cdot x) \right] \tag{7}
$$

$$
= \mathbb{E}_{X \sim p_{\text{data}}, Y \sim p_m, Z \sim p_\gamma, \tilde{X} \sim p_\iota} [\log p_\theta(Y|\tilde{X})]
$$

$$
\leq \mathbb{E}_{X \sim p_{\text{data}}, Y \sim p_m} [\log \mathbb{E}_{Z \sim p_\gamma, \tilde{X} \sim p_\iota} \log p_\theta(Y|\tilde{X})] \,,
$$

where we used Jensen inequality to obtain this last inequality. Thus the L2X objective is a lower bound of the likelihood in the post-hoc setting as the target distribution comes from the fixed classifier $p_m$ we want to explain. The objective described is shed under the light of the Surrogate Post-Hoc regime. By considering $p_m$ as the true data distribution $p_{\text{data}}$, this can be extended to the Free In-Situ regime.

### A.2    INVASE (YOON ET AL., 2018)

We consider a selector $p_\gamma$ (this corresponds to $\pi_\theta(x, s)$ in the original paper) and a classifier $p_\theta$ (this corresponds $f^\phi$ in the original paper) using a 0-imputation to parameterise the imputation module. We consider a regularization function controlled by $\lambda$ such that $R(\gamma) = \mathbb{E}_{p_\gamma}(\|z\|_0)$. They also train a baseline classifier, since this classifier is simply used to reduce variance in the estimated gradient and does not change the objective for the optimization in $\theta$ and $\gamma$.

We can rewrite Eq. (5) in Yoon et al. (2018) using our notation as

$$
\hat{l}(z, x, y) = - \sum_{i=1}^c y_i \log p_\theta(y|x, z) \,. \tag{8}
$$

The loss for the selector network is then

$$
\int_{X,Y} p_{\text{data}}(x, y) \sum_{z \in \{0,1\}^D} [p_\gamma(z|x) \hat{l}(z, x, y) + \lambda \|z\|_0] \mathrm{d}x \mathrm{d}y
$$

$$
= \mathbb{E}_{X,Y \sim p_{\text{data}}} \mathbb{E}_{Z \sim p_\gamma} [\hat{l}(z, x, y) + \lambda \|z\|_0]
$$

$$
= \mathbb{E}_{X,Y \sim p_{\text{data}}} \mathbb{E}_{Z \sim p_\gamma} [- \log p_\theta(Y|X, Z) + \lambda \|z\|_0]
$$

$$
= \mathbb{E}_{X,Y \sim p_{\text{data}}} \mathbb{E}_{Z \sim p_\gamma} [- \log p_\theta(Y|X \cdot Z) + \lambda \|z\|_0]
$$

$$
= \mathbb{E}_{X,Y \sim p_{\text{data}}} \mathbb{E}_{Z \sim p_\gamma} [- \log \mathbb{E}_{\tilde{X}} p_\theta(Y|\tilde{X}) + \lambda \|z\|_0]
$$

$$
= \mathbb{E}_{X,Y \sim p_{\text{data}}} \mathbb{E}_{Z \sim p_\gamma} [- \log \mathbb{E}_{\tilde{X}} p_\theta(Y|\tilde{X})] + \lambda \mathbb{E}_{Z \sim p_\gamma} [\|z\|_0]
$$

$$
= \mathbb{E}_{X,Y \sim p_{\text{data}}} \mathbb{E}_{Z \sim p_\gamma} [- \log \mathbb{E}_{\tilde{X}} p_\theta(Y|\tilde{X})] + \lambda R(\gamma, X)
$$

$$
\geq \mathbb{E}_{X,Y \sim p_{\text{data}}} [- \log \mathbb{E}_{Z \sim p_\gamma} \mathbb{E}_{\tilde{X}} p_\theta(Y|\tilde{X})] + \lambda R(\gamma, X) \,.
$$

Though the loss for the optimisation of the selector with $\gamma$ and the loss for the optimisation in $\theta$ are separated in the original article, they only differ by the regularization $\mathbb{E}_{Z \sim p_\gamma} [\lambda \|z\|_0]$ which is

independent of $\theta$. Thus, minimizing both $\theta$ and $\gamma$ using the loss for the selector network is equivalent to the minimization set up in Invase.

The last equation is exactly the negative log-likelihood of the LEX model with a parametrization described above. Hence the minimisation target of INVASE is a minimisation of an upper bound of the negative log-likelihood.

## A.3 REAL-X (JETHANI ET AL., 2021A)

Let us consider a fixed classifier $p_m$. Similarly to INVASE, we consider a selector $p_\gamma$ (this corresponds to $q_{\text{sel}}$) and a classifier $p_\theta$ (which correspond to $q_{\text{pred}}$) and a function controlled regularizer $R$.

We can rewrite Eq. (3) from Jethani et al. (2021a) in our framework as

$$\mathbb{E}_{x \sim p_{\text{data}}} \mathbb{E}_{y \sim p_m} \mathbb{E}_{z \sim p_\gamma} [\log p_\theta(y|x \cdot z)] + \lambda R(\gamma)$$
$$\leq \mathbb{E}_{x \sim p_{\text{data}}} \mathbb{E}_{y \sim p_m} [\log \mathbb{E}_{z \sim p_\gamma} p_\theta(y|x \cdot z)] + \lambda R(\gamma)$$

The last equation is a lower bound on the log-likelihood of a LEX model parametrized as described above.

As opposed to the previous method, the predictor is trained on a different loss than the selector:

$$\mathbb{E}_{x \sim p_{\text{data}}} \mathbb{E}_{y \sim p_m} \mathbb{E}_{z \sim \mathcal{B}(0.5)} [\log p_\theta(y|x \cdot z)] \,,$$

where $\mathcal{B}(0.5)$ is a distribution of independent Bernoulli for the mask.

This loss is completely independent of the selector's parameters. Thus, training first the predictor network $p_\theta$ until convergence and then training the selector $p_\gamma$ is equivalent to the alternating training in Algorithm 1 in Jethani et al. (2021a).

Thus, we can consider that the associated LEX model is always trained with a fixed $\theta$ and REAL-X is maximizing a negative log-likelihood of a LEX parametrization in a fixed $\theta$ setting.

## B MULTIPLE IMPUTATION SCHEME

### B.1 VAEAC

VAEAC (Ivanov et al., 2018) is a arbitrary conditioned variational autoencoder that allows us to sample from an approximation of the true conditional distribution. To impute a single example, we first sample a latent $h$ according to $p_{prior}$ using a prior network, then sample the unobserved features $p_{gen}$ using the generative network

$$p_\iota(\tilde{x}|x, z) = \int_h p_{prior}(h|x_z, z) p_{gen}(\tilde{x}_{1-z}|x_z, z, h) \mathbf{1}_{\tilde{x}_z = x_z} \,. \tag{9}$$

We only use the VAEAC for the CelebA Smile experiment and leverage the architecture and weights provided in the paper.

### B.2 GAUSSIAN MIXTURE MODEL (GMM)

The Gaussian mixture model allows for fast imputation of masked samples for low to medium size dataset when using spherical Gaussians. Before training, we fit the mixture model to the fully observed dataset by maximizing the log likelihood of the model Eq. (10) using the expectation maximization algorithm. In practice, we use the Gaussian Mixture Model library from Pedregosa et al. (2011) to obtain the parameters $(\pi^k, \mu^k, \Sigma^k)_K$. In the case of spherical Gaussians, $\Sigma_k$ is a diagonal matrix. thus $\forall k, \mu_k, \Sigma_k$ are the size of the input dimension $D$.

$$\mathcal{L}_{\text{GMM}}(x) = \sum_k \pi^k \mathcal{N}(x|\mu^k, \Sigma^k) \,. \tag{10}$$

To obtain a sample imputing the missing variable, we start by calculating $p(k|x, z)$. In the particular case of spherical Gaussian, this allows us to consider only the unmasked features when calculating this quantity.

$$p(k|x,z) = \frac{\mathcal{N}(x_z|(\mu^k)_z,(\Sigma^k)_z)}{\sum_{k'}\mathcal{N}(x_z|(\mu^{k'})_z,(\Sigma^{k'})_z)} \, . \tag{11}$$

Finally, we sample a center $k$ from the previous conditional distribution and a sample from the associated Gaussian:

$$p_\iota(\tilde{x}|x,z) = \sum_k p(k|x,z)\mathcal{N}(\tilde{X}|\mu^k,\Sigma^k) \, . \tag{12}$$

In practice, to train the Gaussian mixture model on discrete image, we add some uniform noise to the input data to help the learning of the input data.

## B.3 Means of Gaussian mixture model (Means of GMM)

We propose an extension of the previous GMM model to get more in distribution samples from the dataset. After sampling a center from the conditional distribution Eq. (11), instead of resampling the imputed values from the Gaussian distribution, we can use directly the means of the centers of the sampled center as imputed data.

$$p_\iota(\tilde{x}|x,z) = \sum_k p(k|x,z)\mathbf{1}_{\tilde{x}=\mu^k} \, . \tag{13}$$

## B.4 Dataset Gaussian Mixture Model (GMM Dataset)

In a second extension of the previous GMM imputation we make use of the validation dataset to create imputations. To that end, we store the quantity $p_{\text{resampling}}(x_{val}^i|k)$ for every center $k$ and every data point $x_{val}^i$ in the validation set, where

$$p_{\text{resampling}}(x_{val}^i|k) = \frac{\mathcal{N}(x_{val}^i|\mu_k,\Sigma_k)}{\sum_j \mathcal{N}(x_{val}^j|\mu_k,\Sigma_k)} \, . \tag{14}$$

After sampling a center from the conditional distribution, we sample one example according to the distribution in Eq. (14).

Hence

$$p_\iota(\tilde{x}|x,z) = \sum_k p(k|x,z) \sum_j p_{\text{resampling}}(x_{\text{val}}^j|k)\mathbf{1}_{\tilde{x}=x_{\text{val}}^j} \, . \tag{15}$$

## B.5 KMeans and Validation Dataset (KMeans Dataset)

By using the KMeans instead of the GMM for the GMM Dataset, we can obtain a simpler multiple imputation methods. To that end, we calculate the minimum distance between a masked input $x_z$ and the masked centers of the clusters $\mu_k$. We select cluster $k*$ closest to the input data and we can sample uniformly any image from the validation dataset belonging to this cluster.

## B.6 Mixture of logistics

We propose to use a mixture of discretized logistic as an approximation for the true conditional distribution. For each center $k$ of the mixture among $K$ centers, we consider a set of $D$ center $\mu_d^k$ and scale parameters $s_d^k$ which allows the creation of a discretized logistic distribution for each pixel similar to the construction in Salimans et al. (2017). We obtain the parameters by maximizing the log-likelihood of the model Eq. (16):

$$p(x) = \sum_k \pi_k \sum_{d\in[0,D]} \text{logistic}(x_d|\mu_d^k,s_d^k) \, , \tag{16}$$

where $\text{logistic}(x_d|\mu_d^k,s_d^k) = \sigma((x_d+0.5-\mu_d^k)/s_d^k)) - \sigma((x_d-0.5-\mu_d^k)/s_d^k))]$ and $\sigma$ is the logistic sigmoid function. In the edge cases of a pixel value equal to 0, we replace $x-0.5$ by $-\infty$ and for 255 we replace $x+0.5$ by $+\infty$.

We initialise the model means and weights by using the $K$-Means algorithm from Sklearn (Pedregosa et al., 2011). We then learn the model either by stochastic gradient ascent on the likelihood or by stochastic EM (both methods leads to similar choice of parameters).

Similarly to the GMM, we sample an imputation by first sampling a center from the mixture using the distribution $p(k|x,z)$, where

$$p(k|x,z) = \frac{\sum_{d\in[0,D]} \text{logistic}(x|\mu_d^k, s_d^k)}{\sum_{k'} \pi_{k'} \sum_{d\in[0,D]} \text{logistic}(x|\mu_d^{k'}, s_d^{k'})} . \tag{17}$$

We can then sample the imputed data using the parameters obtained from the subset restricted sample, that is :

$$p_\iota(\tilde{x}|x,z) = \sum_k p(k|x,z) \sum_{d|z_d=1} \text{logistic}(x_d|\mu_d^k, s_d^k) . \tag{18}$$

### B.7 Means of Mixture of logistics

Instead of sampling from the logistic distribution, after sampling a center $k$ from the conditional distribution Eq. (11), we can use directly the means of the sampled centers as imputed data, that is,

$$p_\iota(\tilde{x}|x,z) = \sum_k p(k|x,z) \sum_d \mathbf{1}_{\tilde{x}_d=\mu_d^k} . \tag{19}$$

### B.8 Validation Dataset (Marginal)

We can sample randomly from the validation dataset to replace the missing value from the unobserved dataset. This corresponds to approximating the conditional true imputation $p_{data}(x_{1-z}|x_z)$ by the unconditional marginal distribution $p_{data}(x_{1-z})$. This type of multiple imputation were already use for

$$p_\iota(\tilde{x}|x,z) = p_{\text{data}}(\tilde{x}_{1-z}) . \tag{20}$$

## C Dataset details

### C.1 Synthetic Dataset generation

The input features for the synthetic datasets follow the generation procedure :

$$\{x_i\}_{i=1}^{11} \sim \mathcal{N}(0,1) \quad y \sim \mathcal{B}\left(\frac{1}{1+f(x)}\right) . \tag{21}$$

We consider different functions $f$ for different situations:

- $f_A(x) = \exp(x_1 x_2)$,
- $f_B(x) = \exp(\sum_{i=3}^6 x_i^2 - 4)$,
- $f_C(x) = \exp(-10\sin(0.2x_7) + |x_8| + x_9 + e^{x-10} - 2.4)$.

This leads to the following datasets:

- **S1**:
$$f(x) = \begin{cases} f_A(x) & \text{if } x_{11} < 0 \\ f_B(x) & \text{if } x_{11} \geq 0. \end{cases} \tag{22}$$

- **S2**:
$$f(x) = \begin{cases} f_A(x) & \text{if } x_{11} < 0 \\ f_C(x) & \text{if } x_{11} \geq 0. \end{cases} \tag{23}$$

- **S3**:
$$f(x) = \begin{cases} f_B(x) & \text{if } x_{11} < 0 \\ f_C(x) & \text{if } x_{11} \geq 0. \end{cases} \tag{24}$$

## C.2 PANEL MNIST AND FASHIONMNIST

Both dataset MMNIST and FashionMNIST are transformed in the same fashion as Jethani et al. (2021a) by dividing the input features by 255. Note that this transformation will affect the choice of the optimal constant of imputation for LEX. We create the train set and validation set of the panel dataset by using only images from the train datasets of MNIST and FashionMNIST and the test set by using only images from the test datasets. The split between train and validation is split randomly with proportion $80\%, 20\%$. Hence, the train dataset of the switching panels input contain 48,000 images, the validation dataset contains 12,000 images and the test dataset 10,000 images.

## C.3 CELEBA SMILE

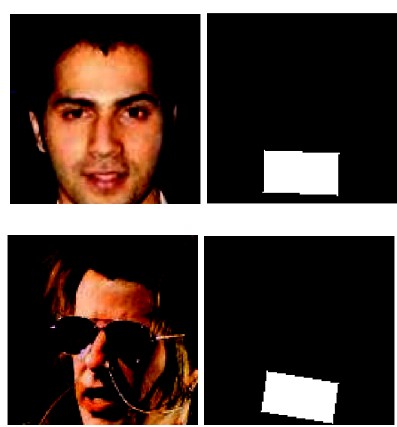

The CelebA dataset consists of $162, 770$ train, $19867$ validation and $19962$ test color images of faces of celebrities of size $178 \times 218$. Before training any model, we crop the image to keep a $128 \times 128$ pixels size square in the center of the image (using Torchvision's *Center-Crop* function and we normalize the channels in datasets. Note that we use this center crop to be able to use directly the weights and parametrization provided by the author of Ivanov et al. (2018) with CelebA. We define the target of this dataset using the attribute given for smiling in the dataset.

We can create the maximum ground truth boxes by using the landmarks position of the mouth. Given the position of both extremities of the mouth, we can obtain both the direction and the lenght of the mouth. Supposing not only the mouth but also the region around is useful for the classifier, we create the maximum ground truth boxes using a rectangle oriented following the direction of the mouth,

Figure 9: Two examples from the CelebA smile dataset associated with their ground truth selection.

centered at the center of both mouth extremities and with height corresponding to two times the lenght of the mouth and width corresponding to the length of the mouth.

## D EXPERIMENT DETAILS

For every experiment, we use an importance weighted lower bound with 10 importance samples for the mask and a single important sample for the imputed values. We estimate the monte carlo gradient using REBAR with the weighted reservoir sampling and the relaxed subset sampling distribution as control variate.

**Synthetic Dataset**    The predictor $p_\theta$ is parameterized with a fully connected neural network $f_\theta$ with 3 hidden layers while the selector $p_\gamma$ is parameterized with a fully connected neural network $g_\gamma$ with 2 hidden layers. The hidden layers are dimension 200 and use ReLU activations. The predictor has a softmax activation for classification while the selector uses a sigmoid activation. We trained all the methods for 1000 epochs using Adam for optimisation with a learning rate $10^{-4}$ and weight decay $10^{-3}$ with a batch size of 1000. Both selector and predictor are free during training for experiment with constant imputation and multiple imputation. When using a surrogate imputation, the surrogate is trained at the same time as the selector according to the algorithm in Jethani et al. (2021a).

**Switching Panels**    The predictor is composed by 2 sequential convolution block that outputs respectively 32, 64 filters. Each block is composed with 2 convolutional layers and an average pooling layer. We fed the output of the last convolutional block to a fully connected layer with a softmax activation. The selector is a U-Net Ronneberger et al. (2015) with 3 down sampling and up sampling blocks and a sigmoid activation. The U-Net outputs a 28x56x1 image mask corresponding to the parameters for each pixel in the image. We trained all the methods for 100 epochs using Adam for optimisation with a learning rate $10^{-4}$ and weight decay $10^{-3}$ with a batch size of 64. Both selector

and predictor are free during training for experiment with constant imputation and multiple imputation. When using a surrogate imputation, the surrogate is trained at the same time as the selector according to the algorithm in Jethani et al. (2021a).

**CelebA**  the predictor neural network $f_\theta$ is composed by 4 sequential convolution block that outputs respectively 32, 64, 128 and 256 filters. Each block is composed with 2 convolutional layers and an average pooling layer. We fed the output of the last convolutional block to a fully connected layer with a softmax activation. The selector is a U-Net (Ronneberger et al., 2015) with 5 down sampling and up sampling blocks and a softmax activation. We lower the number of possible mask by creating a $32 \times 32$ grid of $4 \times 4$ pixels over the whole image. The U-Net outputs a $32 \times 32 \times 1$ image where each feature corresponds to the parametrization of a $4 \times 4$ pixel square in the original image.

For experiments with a constant imputation or a multiple imputation, we train the predictor for 10 epochs. We then train the selector using the pre-trained fixed classifier for 10 epochs. For experiments with a surrogate constant imputation, we train the surrogate for 10 epochs using an independent Bernoulli distribution to mask every pixel (this corresponds to the objective of EVAL-X in Eq (5) in Jethani et al. (2021a)). We then train the selector using the pretrained surrogate for 10 epochs. We optimize every neural netowrk using Adam with a learning rate $10^{-4}$ and weight decay $10^{-3}$ with a batch size of 32

## E   A DIFFICULT OPTIMIZATION PROBLEM

### E.1   AN IMPORTANCE WEIGHTED LOWER BOUND

To maximise Eq. (5), one difficulty resides in the two expectations inside the log. Using Jensen inequality, we can get a lower bound on this log-likelihood.

$$
\begin{aligned}
\mathcal{L}(\theta, \gamma) &= \sum_{n=1}^{N} \log[\mathbb{E}_{Z \sim p_\gamma(.|x^n)} \mathbb{E}_{\tilde{X} \sim p(.|(x^n)Z)} p_\theta(y^n|\tilde{X})] \\
&\geq \sum_{n=1}^{N} \mathbb{E}_{Z \sim p_\gamma(.|x^n)} \log[\mathbb{E}_{\tilde{X} \sim p(.|(x^n)Z)} p_\theta(y^n|\tilde{X})] \\
&\geq \sum_{n=1}^{N} \mathbb{E}_{Z \sim p_\gamma(.|x^n)} \mathbb{E}_{\tilde{X} \sim p(.|(x^n)Z)} \log[p_\theta(y^n|\tilde{X})]
\end{aligned}
$$

This bound may be too loose and give a poor estimate of the likelihood of the model. Instead, we propose to use importance weighted variational inference (IWAE) Burda et al. (2015) so we can have a tighter lower bound than with Jensen inequality. Note that we have to apply two times the IWAE, for the expectation on masks and on the imputed features. For each data point $x_n$, we sample $L$ mask importance samples and $LK$ imputations importance samples.

The IWAE lower of the log-likelihood on the mask with $L$ mask samples

$$
\mathcal{L}(\theta, \gamma) \geq \mathcal{L}_L(\theta, \gamma) \,, \tag{25}
$$

where

$$
\mathcal{L}_L(\theta, \gamma) = \sum_{n=1}^{N} \mathbb{E}_{Z_{n,1}, \dots, Z_{n,L} \sim p_\gamma(.|x_n)} \mathbb{E}_{\tilde{X}_{n,1}, \dots, \tilde{X}_{n,L}, \sim p_\iota(.|Z_{n,l}, x_n)} [\log \frac{1}{L} \sum_{l=1}^{L} p_\theta(y_n|\tilde{X}_{n,l})] \,.
$$

For a fixed $L$ mask imputations, the IWAE lower bound with $K$ imputation samples is

$$
\mathcal{L}_L(\theta, \gamma) \geq \mathcal{L}_{L,K}(\theta, \gamma) \,, \tag{26}
$$

where

$$
\mathcal{L}_{L,K}(\theta, \gamma) = \sum_{n=1}^{N} \mathbb{E}_{Z_{n,1}, \dots, Z_{n,L} \sim p_\gamma(\cdot|x_n)} \mathbb{E}_{\tilde{X}_{n,1,1}, \dots, \tilde{X}_{n,L,K} \sim p_\iota(\cdot|Z_{n,l}, x_n)} \left[ \log \frac{1}{L} \frac{1}{K} \sum_{l=1}^{L} \sum_{k=1}^{K} p_\theta(y_n|\tilde{X}_{n,l,k}) \right] \,.
$$

Theorem 3 of Domke & Sheldon (2018) ensures that when $L \to +\infty$,

$$\mathcal{L}_L(\theta, \gamma) = \mathcal{L}(\theta, \gamma) + O\left(\frac{1}{L}\right),$$

and, for a fixed $L$,

$$\mathcal{L}_{L,K}(\theta, \gamma) = \mathcal{L}_L(\theta, \gamma) + O\left(\frac{1}{K}\right).$$

Thus, using a large number of importance samples for both the mask and the imputation ensures that we are maximizing a tight lower bound of the log-likelihood of the model.

Note that other models falling into the LEX framework are optimizing a lower bound of the log-likelihood (see Appendix A for details) with the Jensen inequality. However, casting their method into the statistical learning framework motivates to choose a tighter lower bound of this log-likelihood which improves results in classification and selection.

### E.2 Gradient Monte Carlo Estimator

We want to train the maximum likelihood model with stochastic gradient descent. Using Monte Carlo gradient estimator for $\theta$ is straightforward. Finding Monte Carlo Gradient estimator for $\gamma$ is more complicated because the expectation on masks depends on $\gamma$ and we sample from a discrete space $\{0, 1\}^D$. A simple way of getting an estimator for this gradient is by using a policy gradient estimator (Sutton et al., 1999) or Score Function Gradient estimator (Mohamed et al., 2020).

On the other hand, by relaxing the discreteness of the distribution, it is possible to reparametrize the expectation in $\gamma$ and use a pathwise monte carlo gradient estimator(Mohamed et al., 2020). These estimators introduce some bias but lower the variance of the estimation. For instance, Yoon et al. (2018) proposed to use the concrete distribution (Maddison et al., 2016) which is a continuous relaxation of the discrete Bernoulli distribution. Similarly, Xie & Ermon (2019), Chen et al. (2018) used some forms of continuous relaxations of subset sampling distribution.

RealX (Jethani et al., 2021a) proposed to use REBAR (Tucker et al., 2019) to further reduce the variance of these gradient estimators while still keeping an unbiased estimator thereof by using the relaxation of the discrete distribution as a control variate for a score function gradient estimator.

When using multiple imputation, there is no possibility to use a continuous mask as the imputation distribution $p_\iota(\tilde{x}|z, x)$ would not be properly defined. To that end, we leverage the allowed reparametrization of the continuous relaxation of the discrete distribution but still apply a straight-through estimator Bengio et al. (2013). When using independent Bernoulli for $p_\gamma$, we consider a thresholded straight through estimator with threshold $t = 0.5$. For relaxed subset sampling, we use a top K function for the straight-through estimator with $k$ being the number of features to be selected. Using this new straight-through estimator for the different continuously relaxed distribution, we can either use a pathwise Monte Carlo gradient estimator or a variation of REBAR where the relaxation is modified by the straight-through function.

## F Extended results

### F.1 Extended results with a fixed sampling rate

**SP-MNIST** In Fig. 11, we observe results for different constants of imputation using a surrogate on SP-MNIST. We see that changing the constant of imputation for the surrogate may drastically change the performance of the selection.

On Fig. 20, we can observe the average of a 100 samples from a model trained with different constant imputation and a surrogate function and an average rate of selection of 5%. We can see that the constant imputation drastically changes the shape of the imputation even though we are using a surrogate function. When using 0, the selection model seems to try to recreate the full image instead of selecting the correct panel. Using constant 1 and 3 seems to recreate the negative of the shape of the two on the correct panel. Finally, when using constant $-1$, the selection recreates the target number in both panels to facilitate classification. These samples using a surrogate function can

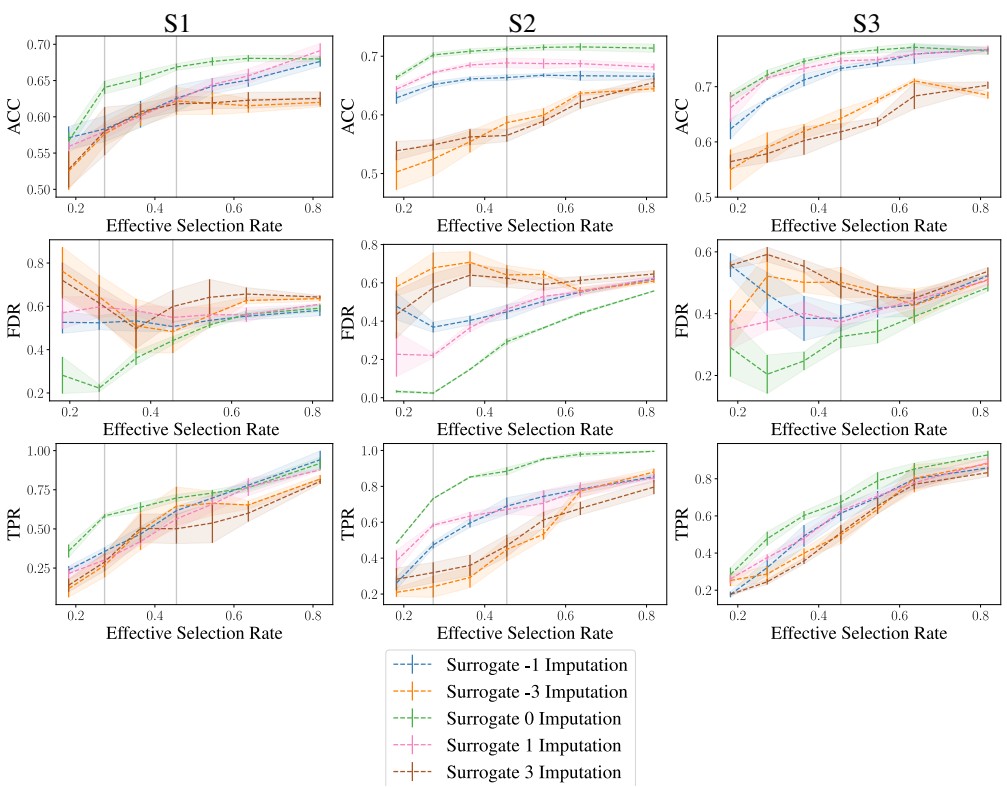

Figure 10: Performances of LEX with different imputation constants using a surrogate constant imputation on the synthetic datasets. This corresponds to the REAL-X parametrization with different constant imputation. Columns corresponds to the three synthetic datasets (S1, S2, S3) and lines corresponds to the different measure of quality of the model (Accuracy, FDR, TPR). [mean ± std over 5 folds/generated dataset]

be considered cheating as the selection is used to improve the classification results. As opposed, samples using multiple imputation on Fig. 21 are less dependent on the type of multiple imputation used.

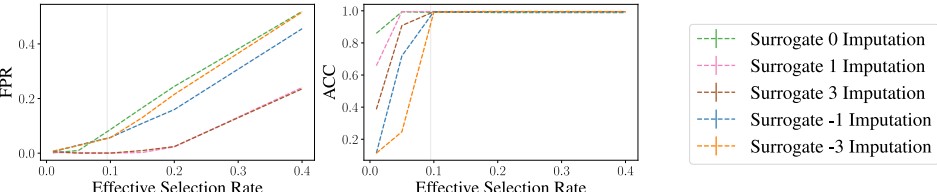

Figure 11: Performance of LEX using a surrogate constant imputation for different imputation constants on the SP-MNIST dataset.

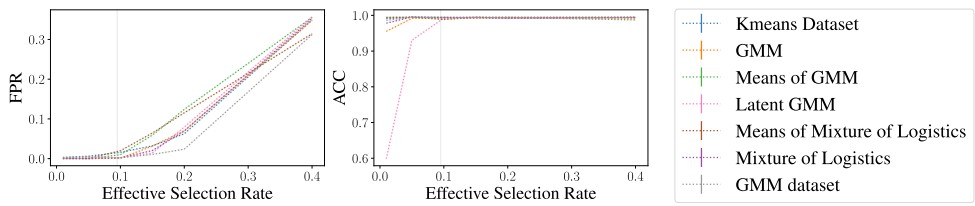

Figure 12: Performances of the LEX method with different methods of multiple imputation on the SP-MNIST dataset.

**SP-FashionMNIST** Following the same procedure as Switching panels MNIST, we consider the dataset Switching panels FashionMNIST. Since on average $50\%$ of the pixels are lit in FashionM-NIST, we expect the true selection rate to be around 25 % of the total number of pixels in Switching Panels FashionMNIST.

In Fig. 13, we compare LEX with two methods of multiple imputation, the Mixture of Logistics and GMM Dataset. These two multiple imputations outperforms their constant imputation counterparts with and without surrogate near the expected true rate of selection. Using the mixture of logistics allows to maintain a strong accuracy, similar to the accuracy of both constant imputation methods. In Fig. 14, we can observe that the selections obtained with the surrogate constant imputation is more robust to the change in constant imputation compared to the Switching Panels MNIST dataset but the variations are still higher that the variations obtain with many different multiple imputation in Fig. 15.

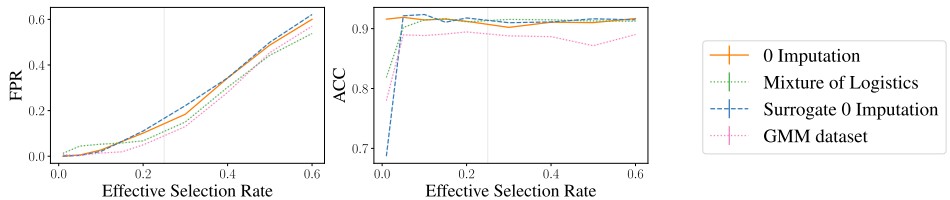

Figure 13: Performances of LEX on the SP-Fashion dataset with different methods of approximation for the true conditional imputation.

**CelebA Smile** On Fig. 16, as opposed to the results on SP-MNIST and the synthetic datasets, we see that the performance of LEX using a surrogate constant imputation does not depend on the choice of a constant.

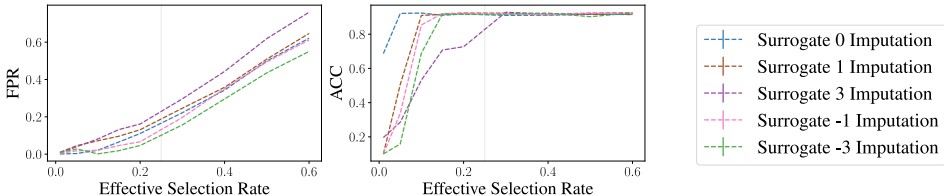

Figure 14: Performance of LEX using a surrogate constant imputation for different imputation constants on the SP-MNIST dataset.

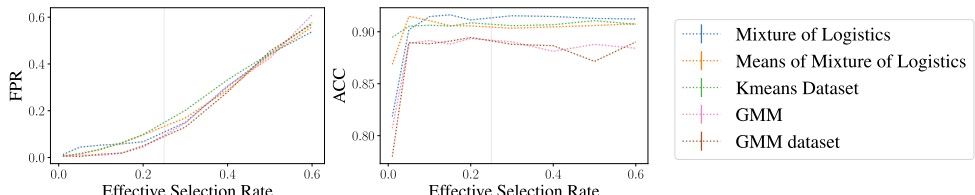

Figure 15: Performances of the LEX method with different methods of multiple imputation on the SP-FashionMNIST dataset.

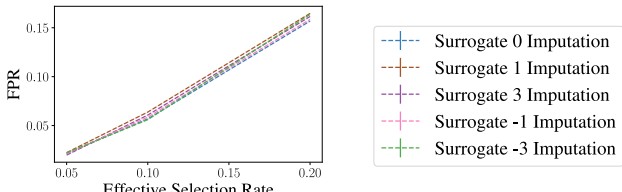

Figure 16: Performances on the CelebA smile dataset with different constant for the surrogate constant imputation.

## F.2 EXTENDED RESULTS USING L1-REGULARIZATION

To provide a better analysis of the influence of the imputation method on the performance of LEX, we fixed all the other parameters hence slightly changing the original methods (Invase and REAL-X). We differed from the original implementations in two ways, the choice of the regularization method and sampling distribution $p_\gamma$ as well as the choice of the Monte-Carlo gradient estimator. Finding an optimal $\lambda$ with L1-regularization is difficult as different ranges of $\lambda$ lead to different performances and different rates of selection depending on the datasets, the imputations (see Appendix F.2.1). While we can estimate an adequate selection rate with intuition from the dataset, no such intuition is available for $\lambda$ which requires a long exploration. In that section, we will study how these choices might affect the performances of LEX and compare to the original implementations using the SP-MNIST dataset.

### F.2.1 ON THE DIFFICULTY OF TUNING $\lambda$

We proposed in Section 4 to study LEX with a fixed selection rate because finding an optimal $\lambda$ requires an extensive search and makes it difficult to compare the results between sets of parameters. In this section, we study how different $\lambda$ lead to very different rates of selection depending on the sets of parameters considered.

We fix the regularization to L1-Regularization and the distribution $p_\gamma$ to be an independent Bernoulli (this is the same setting for distribution and regularization as Invase and REAL-X) and fix the Monte-Carlo gradient estimator to REBAR. We study the evolution of the rate of selection with parameter $\lambda$

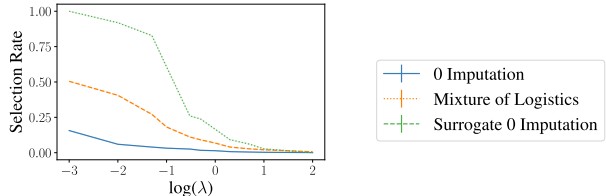

Figure 17: Effective selection rate depending on the value of $\lambda$ of the same LEX model for different types of imputation trained on the SP-MNIST dataset.

varying in $[0.01, 0.05, 0.1, 0.3, 0.5, 1.0, 2.0, 5.0, 10.0, 100.0]$. We observe in Fig. 17 that depending on the imputation, the evolution of the rate of selection is very different. Since we want to compare different methods on the same "credit" of selection (ie the same average rate of selection), we have to search on very large range of $\lambda$ in practice.

### F.2.2 INFLUENCE OF THE MONTE-CARLO GRADIENT ESTIMATION

In their original implementation, L2X, Invase and REAL-X use different Monte-Carlo gradient estimators for the optimization. L2X uses Gumbel-Softmax Maddison et al. (2016) a continuous relaxation of the discrete Bernoulli distribution which is a biased but low variance estimator. Invase uses the REINFORCE Sutton et al. (1999) estimator, an unbiased but high variance estimator. They proposed to control the variance using a baseline network, trained without selection, as a control variate. REAL-X uses the REBAR Tucker et al. (2019) estimator, a REINFORCE estimator using Gumbel-Softmax as a control variate. We chose to focus on REBAR as it can be considered an improvement over REINFORCE and Gumbel-Softmax.

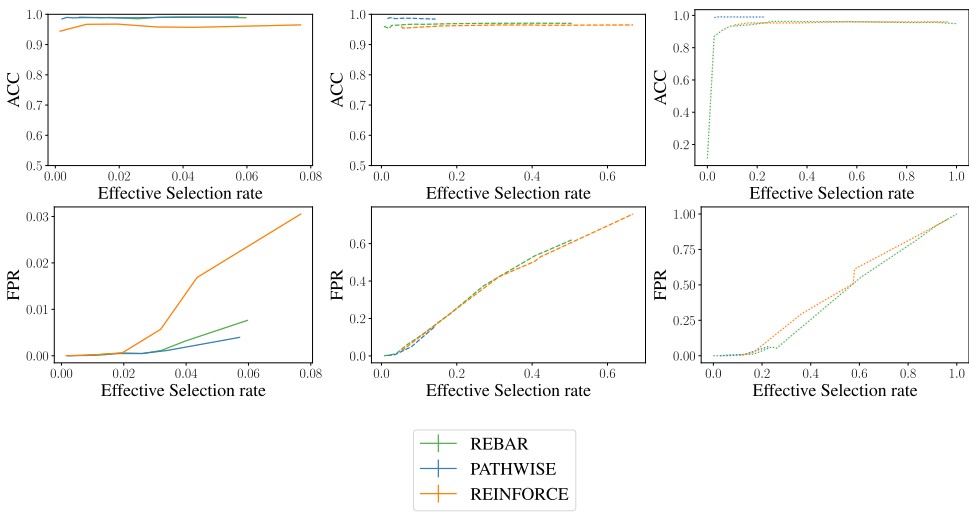

Figure 18: Comparison of the performance of LEX models with L1-regularization with 3 different Monte-Carlo gradient estimator on the SP-MNIST dataset.. $\lambda$ varies in $[0.1, 0.3, 0.5, 1.0, 3.0, 5.0, 10.0,]$. First column corresponds to 0-imputation mimicking the behaviour of INVASE for different MC gradient estimator, second column corresponds to surrogate 0-imputation mimicking the behaviour of REAL-X and third column corresponds to multiple imputation with a Mixture of Logistics.

In this section, we fix the regularization to L1-Regularization and the distribution $p_\gamma$ to be an independent Bernoulli and we compare the difference in performance depending on the Monte-Carlo gradient estimator (REINFORCE, Gumbel-Softmax, REBAR) for different methods of imputation. Note that plain REINFORCE slightly differs from Invase (because of the baseline control variate) but they admitted in the reviews for their paper Yoon et al. (2019) that using the control variate did not improve the results. In Fig. 18, we observe that changing the Monte-Carlo gradient estimator

leads to similar results in prediction and selection. However, changing the Monte-Carlo gradient estimator changes how the choice of $\lambda$ impact the selection rate.

### F.2.3 COMPARISONS WITH THE SET-UPS OF L2X, INVASE AND REAL-X

Here we propose a comparison of the original set-ups of L2X, Invase and REAL-X to two parameterisations of LEX models with multiple imputation. The first one is the same set-up as in 4, using REBAR and an implicit regularization with a fixed rate of selection. For the second set-up, we train using REBAR but consider an Independent Bernoulli for the selection distribution regularized by an explicit L1-Regularisation (note that this is the same regularization and distribution of Invase and REAL-X). $\lambda$ varies in $[0.1, 0.3, 0.5, 1.0, 2.0, 5.0, 10.0]$ for the method with L1-Regularization.

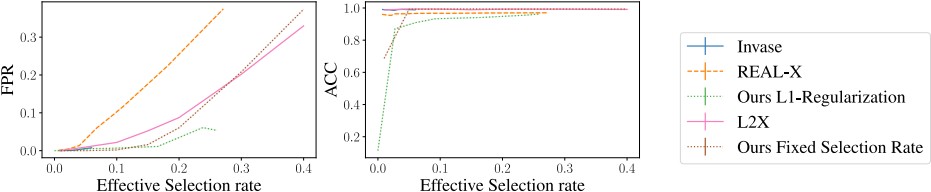

Figure 19: Performances of LEX trained on the SP-MNIST dataset using the same sets of parameters as the original implementation of L2X, Invase, and REAL-X. We compare them to two sets of parameters with Mixture Of Logistics imputation (denoted as Ours) using two types of regularization: L1-Regularization and implicit regularization with a fixed selection rate. The models with L1-regularization were trained on the same grid $\lambda \in [0.1, 0.3, 0.5, 1.0, 2.0, 5.0, 10.0, ]$.

On Fig. 19, we see that our method (i.e. using multiple imputation with a Mixture of Logistics) provides the best performances in the vicinity of the expected selection rate. On the other hand, even with $\lambda$ very close to 0, Invase still selects a very small subset of the features making it difficult to compare to the other methods. Moreover, the accuracy of Invase, REAL-X, and L2X do not decrease with the effective selection rate which suggests that these methods encode the target output in the mask selection.

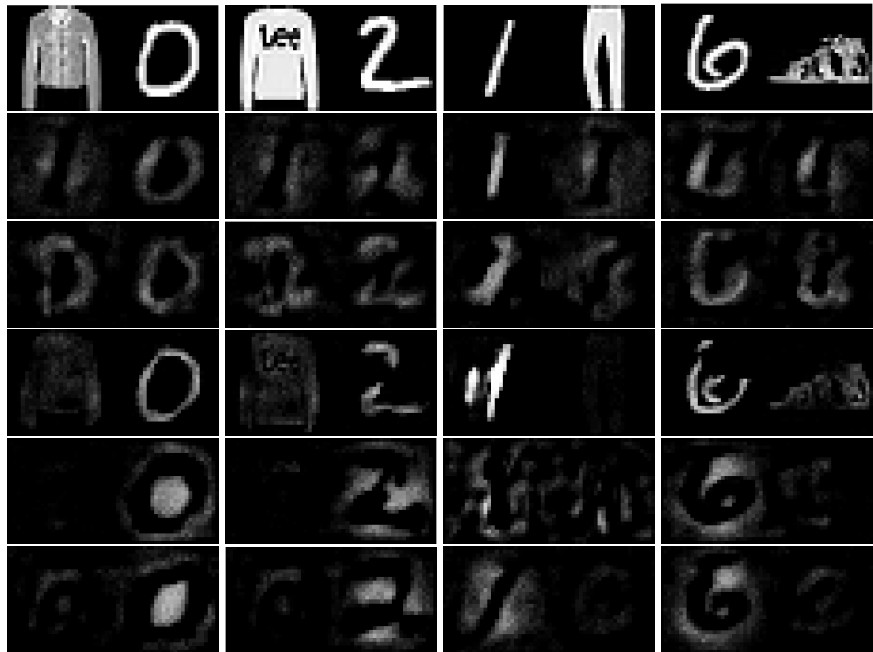

Figure 20: Each figure corresponds to the average of 100 mask samples from the selector trained using a surrogate constant of imputation for different constant. From top to bottom, we have the input data, and constants $-3, -1, 0, 1, 3$. The selector was parametrized by a subset sampling with a rate of selection of 5%.

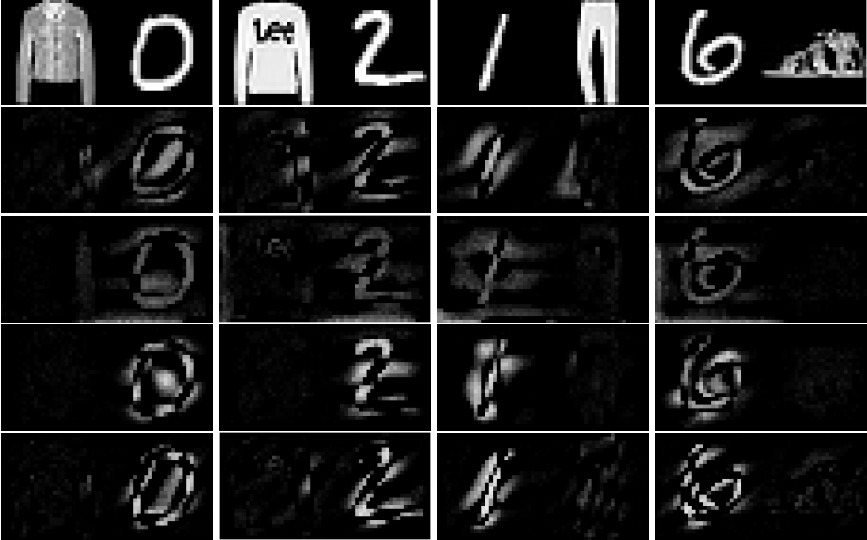

Figure 21: Each figure corresponds to the average of 100 mask samples from the selector trained using different multiple imputation. From top to bottom, the different multiple imputation are : KMeans Dataset, Means of GMM, Mixture of Logitics, means of mixture of logistics. The selector was parametrized by a subset sampling with a rate of selection of 5%.

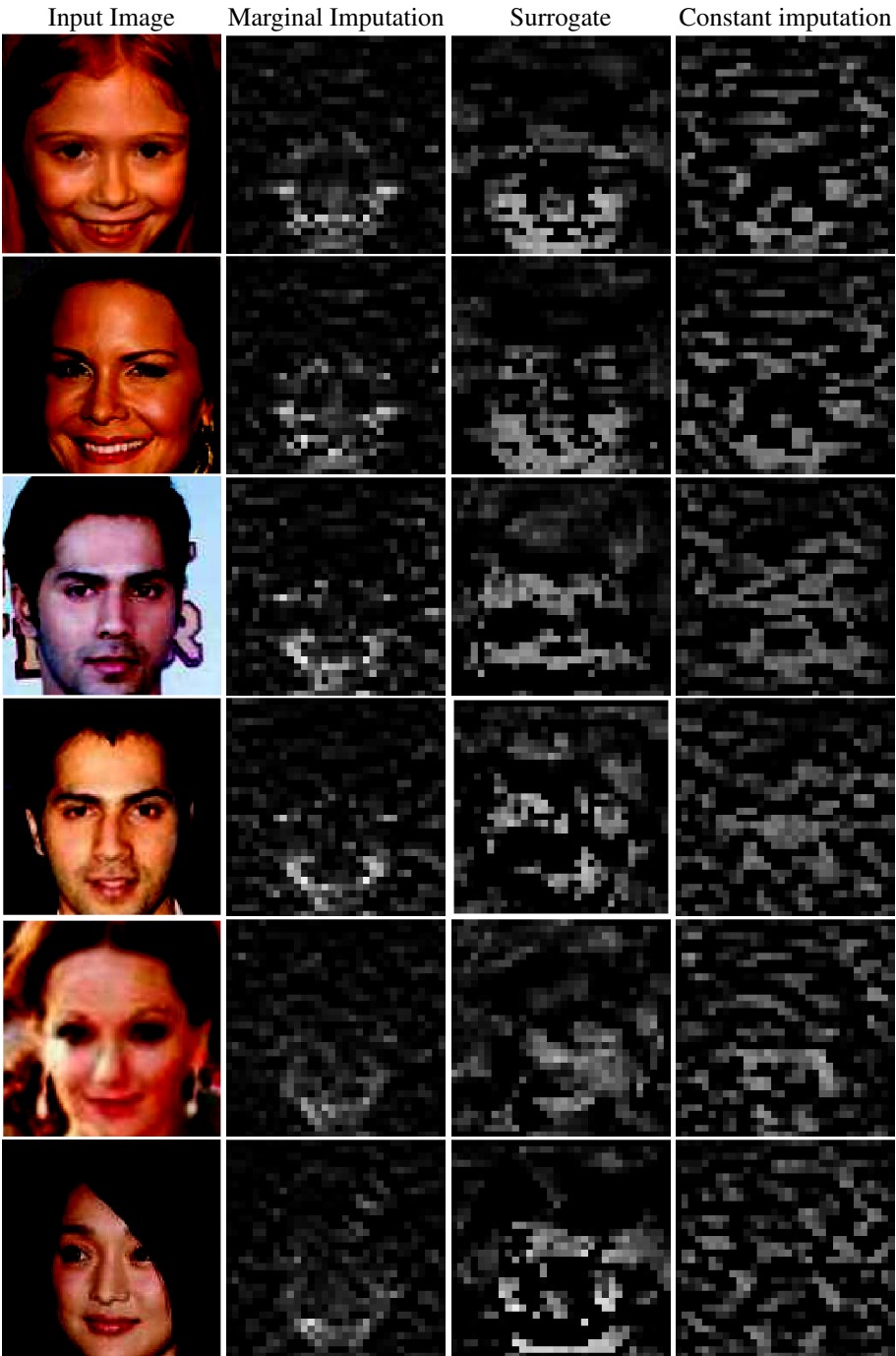

Figure 22: Selection obtained averaging over 100 samples from the distribution $p_\gamma$ parametrized by a subset sampling distribution selecting 10% of the pixels.

