# OpenReview forum: "Explainability as statistical inference"
_ICLR.cc/2023/Conference — Submitted to ICLR 2023_

### Official Review · Reviewer_sxVY · 2022-10-24

**Confidence:** 3
**Correctness:** 3
**Technical Novelty And Significance:** 3
**Empirical Novelty And Significance:** 3
**Recommendation:** 5

**Clarity, Quality, Novelty And Reproducibility:**

- Clarify: as explained above, there are major issues with clarify in the paper.
- Quality: the LEX framework seems technically sound. Too many materials are pushed into the supplementary, and it is a bit challenging to evaluate the other parts of the paper based just on the main paper.
- Novelty: the LEX framework and the results with multiple imputation seems new.
- Reproducibility: seems good.


**Strength And Weaknesses:**

### Strength

- The LEX framework is conceptually appealing, and on a high level is presented in a way that is easy to follow.
- The two new datasets are well-designed and make a lot of sense of evalaution of instance-wise feature-selection methods.
- The observation on the advantage of multiple imputation is valuable and intuitive.

### Weaknesses:

- The main weakness lies in the clarity of the presentation.
    - While on a high level, the LEX framework is presented in a way that is easy to follow, the paper seems to be carelessly written, and is filled with typos, grammar errors, undefined terms and inconsistent notations. A non-exhaustive list of examples include:
      - Figure 1 caption, a standard approaches uses
      - In Section 2.3, why is f_theta is a classifier? It's just a function mapping the input to the parameters of some distributions.
      - In equation 4, why aren't we optimizing the parameter iota? Does the imputation involve any parameters?
      - First sentence of last paragraph in Section 2.4, missing a product in the definition of R?
      - The regularization function R is not consistent between Section 2.4 and Section 2.5. What exactly is the domain of function R?
      - Many terms don't seem to be clearly explained. What is 0 imputation? What is surrogate posthoc?
      - Section 3, data distribution q, why do we call it a data distribution? It seems data distribution should be p_data. And how is q different from p_{\iota}? Why do we change notation here again?
      - What is multiple imputation? Seems like just using a generative model to approximate the true imputation distribution derived from the joint data distribution? Need to be defined in the main text.
      - Section 4 2nd paragraph, want to use of → want to use.
      - Maximum/minimum ground truth selection, used without definition.
      - Section 4 2nd paragraph last sentence, provides → provide
      - Section 4 3rd paragraph, features k s? Typos?
      - InSitu or In-Situ?
      - Section 4 says all the experiments are conducted in the In-Situ regime, but there are two In-Situ regimes according to Section 2.6. What exactly is the setup? Also according to Table 1, several existing methods are not in the In-Situ regime, which seems to suggest the paper is not including any of those methods as baselines.
      - Section 4 mentions a 100 feature importance maps. It's not immediately clear what it means. Also in the next sentence it becomes 100 features instead of 100 feature?
      - Section 4 4th paragraph, three measures, not measure.
      - Section 4.1, selection evaluation, it's not clear what selection rates mean.
    - Too much materials are pushed into the appendix, to the point that it becomes challenging to evaluate the paper based just on the materials in the main text. Some examples include:
        - Section 2.4 talks about different regularization approaches. But which regularization does this paper use exactly?
        - Section 2.5 there should be brief discussions on how existing methods fit into the LEX framework in the main text.
        - Section 3, since multiple imputation is an important point that this paper tries to make, there should at least be brief discussions on what multiple imputation means, what are some of the methods this paper is using in the main text.
        - Section 4.1, just from the main text it's close to impossible to understand what the 3 synthetic datasets look like. There should be at least a brief description. But some of the existing detailed setups can potentially be moved into the appendix.
- Additionally, the way the baselines are set up seems problematic:
    - The paper frames three existing methods (L2X, Invase and REAL-X) under the LEX framework. However, in the experiments the paper only compares multiple imputation with variants in the LEX framework that only loosely correspond to L2X, Invase and REAL-X. For example the paper uses a customized gradient estimator to handle the discrete latent variables, while existing methods adopt other ways of obtaining gradients estimation (e.g. gumbel-softmax and rebar). This makes the comparison difficult to interpret due to multiple moving parts. It would be helpful to also include performance of the original methods in the comparison (e.g. can we use multiple imputation similarly in the original methods? How does that compare with the optimal setup under the LEX framework? How do the original methods perform on the two newly proposed, more complicated datasets?)
- Finally, while the LEX framework is conceptually appealing, the paper does not do a very good job at demonstrating how the framework can benefit the study of instance-wise feature-selection methods. In the experiments the paper mainly demonstrates the benefit of multiple imputation. While intuitive, it seems to me even without the LEX framework we can similarly try to incorporate multiple imputation and study its benefits. It would be beneficial if the authors can more clearly illustrate the benefit of the LEX framework for understanding and improving existing methods. For example, does it lead to insights with which we can design new methods to improve upon existing methods? The lack of fair comparison with existing methods (as explained above) makes it hard to access this point.


**Summary Of The Paper:**

This paper studies amortized instance-wise feature-selection methods. It proposes Latent Variable as Explanation (LEX), a new framework that frames interpretation as a statistical learning method using a unified probabilistic likelihood, and shows how various existing methods can be understood under the LEX framework. The paper proposes two new datasets, a switching panels dataset constructed using MNIST and FashionMNIST, and the CELEBA SMILE dataset, to help evaluate various amortized evaluation methods. Using the proposed LEX framework, the paper demonstrates the advantage of multiple imputation over the combination of constant imputation with surrogate predictors.

**Summary Of The Review:**

While the proposed LEX framework is conceptually appealing, and the newly proposed datasets are well-designed and useful, the paper in its current form is poorly organized and lacks clarity. Additionally, the baseline setups seem problematic. In my opinion the paper is not ready for publication in its current form, but I encourage the authors to improve the writing and organization and include some additional baseline comparisons and resubmit in the future.

------------------

Update after rebuttal: I thank the authors for revising the paper and taking into account my comments. However, after discussing with other reviewers and the AC, we still feel that the paper should either rephrase as presenting the benefits of multiple imputation, or put more efforts into demonstrating the usefulness of the general LEX framework. As a result I am keeping my score.

---

> ### Author Response · Authors · 2022-11-12
> **Answer to Reviewer sxVY (1)**
>
> Many thanks for your comments and assessment of our paper!
>
> # Typos and unclarity corrections
>
> Thank you for the attention with which you read the paper. We corrected the typos you mentioned and some others we found
>
> >"In Section 2.3, why is $f_\theta$ is a classifier? It's just a function mapping the input to the parameters of some distributions."
>
> The prediction problem usually tackled is a classification problem. In that case, the function $f_\theta$ maps the input to the parameters of a Categorical Distribution or a Bernoulli Distribution. Usually, the output of the function $f_\theta$ is exactly the probability to belong to each class hence the name "classifier". However, we do agree that in any other setting than classification, this $f_\theta$ will not be understood as a classifier hence the change in the revised version.
>
> >"In equation 4, why aren't we optimizing the parameter iota? Does the imputation involve any parameters?"
>
> Here $\iota$ is just used to distinguish the distribution $p_{\iota}$ from the other and does not refer to any parameters. During the training of the LEX framework, we suppose that the parameters of the imputation procedure (if any) are not optimized. However, the imputation procedure can contain parameters depending on which imputation procedure you are using. For instance, one may use a VAE to sample multiple imputations but the VAE's parameters will be fixed during the training of LEX.
>
>  > "The regularization function R is not consistent between Section 2.4 and Section 2.5. What exactly is the domain of function R?"
>     \end{quote}
>
> The definition of $R$ was indeed unclear in the article, thank you for pointing it out. We changed it so that the regularization $R$ takes a mask sample $Z$ as input, hoping it will be clearer.
>
> >"Many terms don't seem to be clearly explained. What is 0 imputation? What is surrogate posthoc?"
>
> We added a proper definition for 0-imputation in section 2.3. 0-imputation is a constant imputation scheme with constant $c = 0$. This means that any missing feature is replaced by the value $0$.
> The surrogate post-hoc regime is defined in section 2.6. This is a specific case of training LEX where we want to explain a black-box classifier $p_m$. Instead of directly removing features in the black-box classifier to train the selector, we train a surrogate classifier $p_\theta$ to mimic the behaviour of $p_m$ and the selection at the same time. $p_\theta$ is trained on targets given by $p_m$, hence this is completely equivalent to learning the full self-interpretable model on a dataset generated from $p_{\textrm{data}}(x)p_m(y|x)$.
>
> >"Section 3, data distribution q, why do we call it a data distribution? It seems data distribution should be $p_{data}$. And how is q different from $p_{\iota}$? Why do we change notation here again?"
>
> We wanted to follow the notation presented in Covert et al 2021. but we changed it in the revised version of the article to make it clearer. What we mean by $q$ is indeed an imputation distribution, this has the same role as $p_\iota$.
>
> >"What is multiple imputation? Seems like just using a generative model to approximate the true imputation distribution derived from the joint data distribution? Need to be defined in the main text."
>
> We added a definition of multiple imputation in section 2.3. Single imputation means that for a given instance $x$ and a given mask $z$, there is only a single deterministic reconstruction $\tilde{x}$ (for example 0-imputation is uniquely determined by the mask and the original instance as we just replace the missing values by a constant 0). On the other hand, in multiple imputation, we obtain a distribution $p_{\iota}(\tilde{x}|x_z, z)$ that can handle multiple options of reconstructions. Note that multiple imputation is not necessarily made to approximate the true imputation distribution that is indeed derived from the joint data distribution. Here, we want to approximate this imputation distribution because it is a sensible way of removing features as suggested by Covert et al [1]
> For a review on multiple imputation, we would recommend Murray et al [2]
>
> > "Maximum/minimum ground truth selection, used without definition."
>
> "Maximum ground truth selection" is defined in section 4. The idea is that we don't have access to the complete ground truth selection, but we know a subset of features that should definitely not be selected as it gives no information for the classification. This allows us to measure how wrong the selection is.

---

> > ### Author Response · Authors · 2022-11-12
> > **Answer to Reviewer sxVY (2)**
> >
> > > Section 4 says all the experiments are conducted in the In-Situ regime, but there are two In-Situ regimes according to Section 2.6. What exactly is the setup? Also according to Table 1, several existing methods are not in the In-Situ regime, which seems to suggest the paper is not including any of those methods as baselines.
> >
> > Invase and Real-X are trained in Situ, ie they learn a self-interpretable model directly on the dataset $X,Y \sim p_{data} (X,Y)$. L2X was used only in the Post-Hoc regime in the sense that they first train a classifier $p_m(Y|X)$ on the dataset $X,Y \sim p_{data} (X,Y)$ and then train a self interpretable model $p_{\theta, \gamma}$ that mimics the behaviour of $p_m$ (which is exactly the same as training on the dataset $X,Y \sim p_{data}(X) p_m(Y|X)$). Due to the similarity between Post-Hoc and In-Situ, and since our ground truth is only defined on $p_{data}(Y|X)$ but not on $p_m$, we avoid using the extra classifier and directly trained the self-interpretable model on the dataset. This is still the same setup as L2X but this allows the use of the ground truths for evaluation.
> >
> > > Section 4 mentions a 100 feature importance maps.
> >
> > Thanks for pointing out a mix-up in the denomination. The feature importance map (how important is each feature for the classification) is given by $\mathbb{E}_{\gamma}[Z|X=x]$ which is the average over all the likely masks that are actually used for prediction. To estimate a single feature importance map, we average over a 100 mask samples from $p_\gamma$.
> >
> > >Section 4.1, selection evaluation, it's not clear what selection rates mean.
> >
> > For a given mask, the selection rate is the ratio of the selected features over the number of features $\frac{Card\{z=1\}}{D}$. When using the subset sampling parametrization, this quantity is fixed for all instances while it can vary between instances when using Bernoulli and L1 Regularisation parametrization.
> >
> > # Appendix
> >
> > > "Too much materials are pushed into the appendix
> >
> > Thank you for pointing out the issue with our supplementary. We tried to address many of the concerns so that the paper could suffice by itself but we are still limited by the 9 pages limit.
> > For instance, though we agree it would be clearer for any reader if these synthetic datasets were properly detailed in the main paper, we are limited in space and these papers were extensively presented in the three other papers (L2X, INVASE and REAL-X).
> >
> > # Experiments
> >
> > >  However, in the experiments the paper only compares multiple imputation with variants in the LEX framework that only loosely correspond to L2X, Invase and REAL-X. For example the paper uses a customized gradient estimator to handle the discrete latent variables, while existing methods adopt other ways of obtaining gradients estimation (e.g. gumbel-softmax and rebar).
> >
> > Many different sets of parameters can be chosen under the LEX framework. Here we wanted to focus on the imputation as this was the main focus of many recent papers [1], [3]. To that end, we fixed the other parameters to what we consider optimal choices or choices that facilitates the comparison between sets of parameters. The variants studied in the article differ from their original counterparts (L2X, Invase, REAL-X) in two ways : the Monte-Carlo gradient estimator and the regularization method.
> >
> > Three different Monte-Carlo gradient estimators were used in the original implementation. L2X uses a Gumbel-Softmax estimator,  Invase uses a REINFORCE estimator.  Finally, REAL-X uses REBAR, a REINFORCE estimator whose variance is mitigated by using the Gumbel Softmax continuous relaxation as a control variate. REBAR is known to provide low variance and unbias estimation of the gradient without the need to train an extra baseline predictor. REBAR is considered state of the art for discrete space optimization and thus is an optimal choice for gradient estimation. The customized gradient estimator (with Straight-Through estimator) is only used for multiple imputation and thus does not appear in the baseline we are comparing to. We added some experiments to compare the different Monte-Carlo gradient estimator in Appendix F.2.2.
> >
> > Invase and REAL-X use an explicit L1-regularization controlled by an hyperparameter $\lambda$ whereas we use an implicit regularization by constraining the selection to a fixed number of features. Finding an optimal $\lambda$ was very difficult as the meaningful range (outside which either everything is selected or nothing is selected) varies with datasets and imputation methods.  We added some experiments to show how the selection rate behaves with $\lambda$ when using L1-regularization in Appendix F.2.1. We think more interesting to compare method when they use the same "credit" to select features (i.e. the same rate of selection).
> >
> > However, we added experiments using the same sets of parameters as L2X, Invase and REAL-X in Appendix F.2.3. Thank you for the suggestion.

---

> > > ### Author Response · Authors · 2022-11-12
> > > **Answer to Reviewer sxVY (3)**
> > >
> > >
> > > > It would be beneficial if the authors can more clearly illustrate the benefit of the LEX framework for understanding and improving existing methods.
> > >
> > >
> > > The LEX framework provides a better understanding of the already existing other methods. Indeed, all the other methods considered the selector and predictor as two disjoint parts that worked separately. In their setting, for a single mask selection, the predictor gives a single prediction that is used for evaluating the accuracy of the model. Through LEX, we get a better understanding of how the model handle the multiple mask samples to make a prediction : by averaging the output over all the mask possibilities.
> > >
> > > While training, this gives the intuition for using a bound tighter to the original likelihood ( Appendix E.1). LEX allows for a sounder way to evaluate model performances between different sets of parameters. The prediction output should not be calculated by a single mask sample but by the average over all the masks. Indeed, estimating the feature importance map, L2X proposed a heuristic that does not represent well the underlying reality of what the model is learning : they consider as selected features the features that have the top-k largest score at the output of the network. This type of heuristics hides what the predictor really "sees" when training. For instance, if the weights at the output of the selector network $g_\gamma$ are close to each other, this heuristic does not represent well the importance map.
> > >
> > > The advantages for modelization are multiple. The LEX framework allows for more complicated distributions of mask. One could consider distributions where the selected features are not independent to each other (for instance a selection where neighbouring pixels have more incentive to be selected together). Through our framework, we can leverage the traditional statistical learning theory and come up with other methods for optimizing the parameters. By adding prior on the weights of the network, the LEX framework allows to perform Bayesian inference.
> > >
> > >
> > > [1] COVERT, Ian, LUNDBERG, Scott M., et LEE, Su-In. Explaining by Removing: A Unified Framework for Model Explanation. J. Mach. Learn. Res., 2021, vol. 22, p. 209:1-209:90.
> > >
> > > [2] Jared S Murray. Multiple imputation: a review of practical and theoreticalfindings. Statistical Science, 33(2):142–159, 2018.
> > >
> > > [3] Neil Jethani, Mukund Sudarshan, Yindalon Aphinyanaphongs, and Rajesh Ranganath. Have we learned to explain?: How interpretability methods can learn to encode predictions in their interpretations. In International. Conference on Artificial Intelligence and Statistics, pages 1459–1467. PMLR, 2021.

---

### Official Review · Reviewer_DxL9 · 2022-10-27

**Confidence:** 3
**Correctness:** 4
**Technical Novelty And Significance:** 3
**Empirical Novelty And Significance:** 3
**Recommendation:** 6

**Clarity, Quality, Novelty And Reproducibility:**

The work is strikes me as novel and original. The paper was very
clearly written and straightforward to follow. The only change I would
want is greater clarification of the contributions of the paper in the
introduction.

The experiments are well thought out and while I still feel the
datasets introduced are fairly synthetic they are much more realistic
than previous datasets used in the literature.

While no code was included, the appendix provided enough information
that work seems reproducible.

**Strength And Weaknesses:**

The paper is clearly written and makes a real contribution. The
framework outlined really does seem to add flexibility by allowing for
a more generic imputation strategy to be used. I am curious, why
couldn't the existing methods be adapted to just use multiple
imputations?

My main concern is that the main benefit is the ability to use
amortized imputation methods than necessarily any other portion of the
framework. This comes across a bit incremental, as all other methods
trivially allow changing the selector network and predictor network.

I have some quibbles with the title as it's hardly the first paper that
learned an explainable model as a statistical inference task, but perhaps
this is best cleared up with clarifications of the ways this paper is statistical
inference and Chen et al are not.

It is mentioned that selector when using LEX should still end up encoding the predictor,
but I didn't really see any experiments demonstrating that.

Update: given the contribution seems to be the framework, I think the paper would greatly benefit from some rewriting to better highlight the strengths of that framework. Though I should stress I think the imputation methods discussed in this work are very interesting and would be a great contribution on their own.

**Summary Of The Paper:**

This paper introduces a new method for jointly learning a predictor and an interpretability model. The method
offers a general framework that has LEX, INVASE, and REAL-X as special cases. The paper also introduces two more
datasets synthetically created from FashionMNIST and CelebA with ground truth selected features.


**Summary Of The Review:**

 Interesting and significant contribution to the interpretability
 literature. Paper would benefit from clarifying its unique
 contributions.

---

> ### Author Response · Authors · 2022-11-12
> **Answer to Reviewer DxL9**
>
> Many thanks for your comments and assessment of our paper!
>
>
> > "I am curious, why couldn't the existing methods be adapted to just use multiple imputations? My main concern is that the main benefit is the ability to use amortized imputation methods than necessarily any other portion of the framework. This comes across a bit incremental, as all other methods trivially allow changing the selector network and predictor network"
>
> The other existing methods (L2X, Invase, and REAL-X) are actually LEX models and could use multiple imputations to remove features. However, the way we use multiple imputations is only defined through the lens of the LEX framework. The main difference is that all the other methods presented the predictor and selector as two separate entities not part of a single probabilistic model. In their method, one prediction output is obtained through a single mask sample. When using multiple imputation through their framework, it is not clear whether one should consider a prediction using a single mask and unique sample of imputation or through the average. The LEX framework allows to answer naturally.
>
> Though multiple imputation feels natural within the LEX framework, it is not straightforward. For instance, this required having discrete masks (while a continuous relaxation is still usable with constant imputation) hence the use of the Straight-Through estimator and using a tight lower bound (the Importance Weighted Lower Bound in Appendix E.1).
>
> > "I have some quibbles with the title as it's hardly the first paper that learned an explainable model as a statistical inference task, but perhaps this is best cleared up with clarifications of the ways this paper is statistical inference and Chen et al are not."
>     \end{quote}
>
> Chen et al [1] saw interpretation as an information theoretic problem. They are implicitly maximizing a lower bound of a log-likelihood but on evaluation consider the prediction and selection as two separate task. We explicit the underlying probabilistic model and the likelihood optimized with the cost function they proposed.
>
>
>  >"It is mentioned that selector when using LEX should still end up encoding the predictor, but I didn't really see any experiments demonstrating that."
>
>  We don't have any guarantee that LEX would not encode the prediction. We showed empirically using the newly created datasets that using multiple imputation is less likely to encode the prediction in the selection compare to other imputation methods.
>
> Indeed, the main argument for REAL-X's surrogate constant imputation was that it prevented encoding. We show on the Panel MNIST dataset that using a surrogate constant imputation could still lead to encoding. For instance, we can see obvious encoding in Fig. 20 (second column, third line) as the selector tries to recreate a $2$ in both quadrant. We came up with the panel dataset and their maximum selection ground truth to have a way to measure this "encoding" when using the imputation. If the selector selects pixels in the wrong quadrant, the selector considers as important for the classification part of the image that should not be taken into account and the prediction is therefore leveraging some information from the mask.
>
> > "While no code was included, the appendix provided enough information that work seems reproducible."
>
> We provided the code in the supplementary material with a README.txt on which script to launch to recreate the experiments.
>
>  [1] Chen, J., Song, L., Wainwright, M., & Jordan, M. (2018, July). Learning to explain: An information-theoretic perspective on model interpretation. In International Conference on Machine Learning (pp. 883-892). PMLR.

---

### Official Review · Reviewer_Hv7W · 2022-11-04

**Confidence:** 2
**Correctness:** 3
**Technical Novelty And Significance:** 3
**Empirical Novelty And Significance:** 2
**Recommendation:** 6

**Clarity, Quality, Novelty And Reproducibility:**

The paper is clearly written and easy to follow. The method is well-motivated. However, there are a number of typos and undefined terms and symbols. There is no source code provided to directly reproduce the results.

**Strength And Weaknesses:**

Strengths: The proposed method offers a general framework for casting the interpretability problem as a statistical inference problem, encompassing existing methods such as L2X, Invase, and REAL-X. In addition, the insights on using multiple imputation in the proposed procedure could be useful to others.

Weaknesses: The evaluation is not sufficiently convincing. The paper mainly demonstrates the effectiveness of multiple imputation over other imputation variants under the LEX framework, instead of directly comparing LEX with the other existing methods. Moreover, the technical contribution of their proposed method over existing ones has not been stated clearly in the paper. For example, though it does provides a unified framework, it is not the first paper framing the interpretability problem as a statistical inference problem.


**Summary Of The Paper:**

The paper proposes to frame the model interpretability problem as a statistical inference problem and develops a general probabilistic model that trains a predictor and a selector network via maximum likelihood to produce interpretable predictions. The proposed framework LEX is modular and able to encompass existing models including L2X, Invase, and REAL-X. The paper also introduces two datasets with ground truth selection for evaluating the proposed method.

**Summary Of The Review:**

This paper proposes a unified probabilistic framework to solve the interpretability problem as a statistical inference problem, which could be inspiring to the community. However, the authors need to clarify their main contribution and justify why the current baseline comparison experiments are sufficient to demonstrate the proposed method's effectiveness.

---

> ### Author Response · Authors · 2022-11-12
> **Answer to reviewer Hv7W**
>
> Many thanks for your comments and assessment of our paper!
>
> >"Weaknesses: The evaluation is not sufficiently convincing. The paper mainly demonstrates the effectiveness of multiple imputation over other imputation variants under the LEX framework, instead of directly comparing LEX with the other existing methods."
>
> There are a lot of different sets of parameters to choose from for a LEX model. We are mostly interested in the imputation method for removing features as this was the main topic of Jethani et al [1]. To facilitate the comparisons between the imputation methods, we fix all the other parameters to what we consider optimal. The two main differences between the baseline and the original method : the Monte Carlo gradient estimation and the regularization.
>
> L2X uses Gumbel-Softmax (a relaxation of the Bernoulli distribution) to obtain estimations of the monte-carlo gradient. This allows for low variance but biased estimations of the gradient. Invase uses a REINFORCE gradient estimator which is unbiased but has a high variance. To minimize the variance of the estimator, they use a control variate by training a baseline network on the prediction. We instead chose to use REBAR for all variants of LEX, like REAL-X, since REBAR is considered state-of-the-art for optimization in discrete space and should be considered an improvement over the other methods. This is a combination of a REINFORCE estimate and a GumbelSoftmax control variate which allows for unbiased estimation and low variance. We added extra experiments in Appendix F.2.2 of the revised version of the paper to show how different choices of Monte-Carlo gradient estimators affect the performance.
>
> REAL-X and INVASE use an explicit L1-Regularization controlled by a parameter $\lambda$ while we use an implicit regularization by enforcing the selection of a fixed number of features. We found the procedure to obtain the optimal $\lambda$ with such a regularization very complicated as for different sets of parameters, the interesting range of $\lambda$ varies. Indeed, for the same magnitude of $\lambda$, different sets of parameters will lead to very different rates of selection. (we added experiments in Appendix F.2.1 to show this phenomenon). Since we want to compare all methods on both accuracy and selection for the same "credit" of selection, it makes sense to constrain the mask distribution to a fixed number of features. In the original papers, they consider a large grid search and consider as optimal the $\lambda$ that allows for the minimum selection while maintaining $95\%$ accuracy.
>
> We added some experiments using the exact sets of parameters of L2X, Invase, and REAL-X in Appendix F.2.3.
>
>
>
> > "Moreover, the technical contribution of their proposed method over existing ones has not been stated clearly in the paper. For example, though it does provides a unified framework, it is not the first paper framing the interpretability problem as a statistical inference problem."
>
> To the best of our knowledge, we are the first to frame an interpretability method using the two components (selector and predictor) as the maximization of the log-likelihood of a probabilistic model. The original description of L2X, RealX and Invase consider the explanations and the predictions as two separate objectives. However, they still optimize implicitly a lower bound of the likelihood of this model.
>
> Through the lens of this unified framework, we can properly understand how selection and prediction interact (the output prediction is the average over all masks), we can motivate properly the use of multiple imputations (the output prediction is the average over all samples and masks in that case) and propose a sensible way to obtain the importance feature map for any distributions (as the expectation on the mask samples).
>
> Though multiple imputation feels natural with the LEX framework, it is not straightforward either, it requires the use of a tighter lower bound of the log-likelihood (the importance weighted lower bound in Appendix E.1.) and constraining the mask to be discrete hence the use of a Straight-Through estimator when using multiple imputations (Appendix E.2).
>
>  > "However, there are a number of typos and undefined terms and symbols. There is no source code provided to directly reproduce the results."
>
> We have corrected many of the typos and undefined terms in the revised version of the paper. The code is provided in the supplementary material. We added a README file to explain how to reproduce the experiments in the paper.
>
> [1] Neil Jethani, Mukund Sudarshan, Yindalon Aphinyanaphongs, and Rajesh Ranganath. Have we learned to explain?: How interpretability methods can learn to encode predictions in their interpretations. In International. Conference on Artificial Intelligence and Statistics, pages 1459–1467. PMLR, 2021.

---

### Author Response · Authors · 2022-11-12
**General Comments**

We thank the reviewers for their valuable feedback and suggestions. All of them found the LEX framework quite compelling, and appreciated it was “well-motivated” (Reviewer Hv7W), “conceptually appealing” (Reviewer sxVY) and an “interesting and significant contribution to the interpretability literature” (Reviewer DxL9). They also praised the new datasets introduced in the paper as “much more realistic than previous datasets” (Reviewer DxL9) and “well-designed and make a lot of sense” (Reviewer sxVY).
All three reviewers agreed that the paper was “clearly written and easy to follow” (Reviewer Hv7W), “clearly written” (Reviewer DxL9), “on a high level is presented in a way that is easy to follow” (Reviewer sxVY), even though Reviewer Hv7W and Reviewer sxVY they spotted some typos, and encouraged us to clarify some things. As we detail below, we have revised the paper accordingly. Beyond these clarity improvements for which we thank all reviewers, we have also revised the paper in order to include additional baselines, as suggested by Reviewer Hv7W and Reviewer sxVY.

* Reviewer sxVY and Reviewer Hv7W noticed some typos, inconsistencies in the notation and some unclear definitions. We thank them for pointing out these issues and have addressed them all in the revised version on OpenReview.

* Reviewer Hv7W and Reviewer DxL9 pointed out that the code did not provide a way to directly reproduce the results. We added a README.txt file in the code in the supplementary material to reproduce the experiments.


* All three reviewers wanted some clarifications of the contribution as opposed to already existing methods. We have edited the introduction as suggested by Reviewer DxL9 to make the contributions clearer. Our contributions are as follows :
1. We propose \textbf{LEX (Latent Variable as Explanation)} a self-interpretable probabilistic model that allows for feature importance selection. (Section 2)

2. We show that up to different optimization procedures, other existing amortized explanation models (L2X, INVASE, REAL-X) optimize such a likelihood and can thus be cast as part of the LEX framework. (Section 2.5 and Appendix A.)

3. Through the LEX framework, we are able to unify post-hoc (training to explain a classifier) and in-situ training (training a full self-interpretable model). (Section 2.6)

4. LEX is composed of several modules that can be adapted, including an imputation part. We experimentally stress the benefits of using multiple imputations to obtain more plausible explanations. (Section 3 and 4)

5. We propose two new datasets that allow evaluating the selection with a ground truth explanation, i.e., for each instance of data, the associated ground truth defines a subset of features that should not be selected. (Section 4).

* Reviewer Hv7W and Reviewer sxVY wondered why we chose the baselines in the paper. There are many different parameters to select for LEX models but we are mostly interested in the imputation method for removing features as this was the main topic of Jethani et al. [1]. For comparison sake, we fix all the other parameters to choices that we consider optimal or that facilitate the comparison between different parametrizations. There are two main differences between the way the previously existing methods (L2X, INVASE and REAL-X) were implemented and the baselines we compare us to in the paper: the Monte-Carlo gradient estimator and the regularization. Following suggestions from Reviewer sxVY and Hv7z, we added extra experiments using the exact sets of parameters of the original methods in Appendix F.2.3., we added an analysis of the performance depending on the Monte-Carlo gradient estimator in Appendix F.2.2  and on $\lambda$ in Appendix F.2.1.


We thank the reviewers again for the good quality of your comments and remarks and look forward to any additional questions or suggestions.

[1] Neil Jethani, Mukund Sudarshan, Yindalon Aphinyanaphongs, and Rajesh Ranganath. Have we learned to explain?: How interpretability methods can learn to encode predictions in their interpretations. In International. Conference on Artificial Intelligence and Statistics, pages 1459–1467. PMLR, 2021.

---

### Author Response · Authors · 2022-12-11
**General comments after rebuttal**

We would like to thank the reviewers for their thoughtful comments and suggestions. Both reviewers DxL9 and sxVY mentioned some unclarities regarding the benefits from either multiple imputation and the LEX framework as two distinct advantages. The two benefits should be regarded as highly intertwined.
The main advantage of the LEX framework is its modularity, which allows us to compare existing methods under a common framework. This modularity enables us to isolate the different building blocks of such models and search for an optimal parametrization. This is how we identified the imputation step as a potential limitation of existing methods. By using multiple imputation, we can overcome this limitation and improve the interpretation of the model on existing and newly proposed datasets.
We hope this clarifies the benefits of combining multiple imputation and the LEX framework as a single, coherent paper. Thank you again for your feedback and suggestions.

---

### Decision · Program_Chairs · 2023-01-20

**Decision:**

Reject

**Justification For Why Not Higher Score:**

Weaknesses: The utility of the general framework is unclear and needs more discussion to appreciate the contribution of the framework.

**Justification For Why Not Lower Score:**

N/A

**Metareview: Summary, Strengths And Weaknesses:**

The goal of this paper was to unify several methods for learning-to-explain like L2X and REAL-X under a single objective that's modular and directly leads to instancewise explain abilit. The modular pieces are 1) how to turn off features, 2) the model, 3) the selections, and 4) a regularizer for the selections. They use their framework and study a couple of exemplar problems in the field along with a few imaging datasets. The reviewers were initially split on this paper with one positive, one in the middle, and one negative. This was a borderline paper. There was a live meeting between the AC where a common issue emerged in that the reviewers appreciated that while the study of imputation/masking techniques provided more practical guidance, the value of the framework itself was not clear. There was some discussion on whether the paper could be rewritten to focus on practical guidance for imputation and masking, but the reviewers felt that the change would be too large and the paper would need to be re-reviewed.


Strengths: The paper tries to unify existing explainability methods under the view of inference. With this view, different imputation/masking strategies are considered.

Weaknesses: The utility of the general framework is unclear and needs more discussion to appreciate the contribution of the framework.

**Summary Of Ac-Reviewer Meeting:**

There was a live meeting between the AC where a common issue emerged in that the reviewers appreciated that while the study of imputation/masking techniques provided more practical guidance, the value of the framework itself was not clear. There was some discussion on whether the paper could be rewritten to focus on practical guidance for imputation and masking, but the reviewers felt that the change would be too large and the paper would need to be re-reviewed.